



# Disparities in particulate matter (PM₁₀) origins and oxidative potential at a city-scale (Grenoble, France) - Part I: Source apportionment at three neighbouring sites

Lucille Joanna S. Borlaza[1], Samuël Weber[1], Gaëlle Uzu[1*], Véronique Jacob[1], Trishalee Cañete[1], Olivier Favez[2,3], Steve Micallef[4], Cécile Trébuchon[4], Rémy Slama[5], and Jean-Luc Jaffrezo[1]

[1]University Grenoble Alpes, CNRS, IRD, INP-G, IGE (UMR 5001), 38000 Grenoble, France
[2]INERIS, Parc Technologique Alata, BP 2, 60550 Verneuil-en-Halatte, France
[3]Laboratoire Central de Surveillance de la Qualité de l'Air (LCSQA), 60550 Verneuil-en-Halatte, France
[4]Atmo Auvergne-Rhônes Alpes, 38400 Grenoble, France
[5]IAB, Team of Environmental Epidemiology applied to Reproduction and Respiratory Health, University of Grenoble Alpes, 38000 Grenoble, France

*Correspondence to*: Gaëlle Uzu (gaelle.uzu@ird.fr)

**Abstract.** A fine-scale source apportionment of PM₁₀ was conducted in three different urban sites (background, hyper-center, and peri-urban) within 15 km of the city in Grenoble, France using Positive Matrix Factorization (PMF 5.0) on measured chemical species from collected filters (24-hr) from February 2017 to March 2018. To improve the PMF solution, several new organic tracers (3-MBTCA, pinic acid, phthalic acid, MSA, and cellulose) were additionally used in order to identify sources that are commonly unresolved by classic PMF methodologies. An 11-factor solution was obtained in all sites including commonly identified sources from primary traffic, nitrate-rich, sulfate-rich, industrial, biomass burning, aged sea salt, sea/road salt, and mineral dust, and the newly found sources from primary biogenic, secondary biogenic oxidation, and MSA-rich. Generally, the chemical species exhibiting similar temporal trends and strong correlations showed uniformly distributed emission sources in the Grenoble basin. The improved PMF model was able to obtain and differentiate chemical profiles of specific sources even at high proximity of receptor locations confirming its applicability in a fine-scale resolution. In order to test the similarities between the PMF-resolved sources, the Pearson distance and standardized identity distance (PD-SID) of the factors in each site were compared. The PD-SID metric determined homogeneous sources (biomass burning, primary traffic, nitrate-rich, sulfate-rich, primary biogenic, MSA-rich, aged sea salt, and secondary biogenic oxidation) and heterogeneous sources (industrial, mineral dust, and sea/road salt) across different urban sites, thereby allowing to better discriminate localized characteristics of specific sources. Overall, the addition of the new tracers allowed the identification of substantial sources (especially in the SOA fraction) that would not have been identified or possibly mixed with other factors, resulting in an enhanced resolution and sound source profile of urban air quality at a city scale.



# 1 Introduction

Atmospheric aerosols, or particulate matter (PM), are complex mixtures of particles from direct and indirect emissions (e.g., gas-to-particle conversion processes) that are from natural and anthropogenic sources in the atmosphere (Wilson and Spengler, 1996). The growing interest in ambient aerosol studies is driven by their impacts on health, air quality, and global climate (Colette et al., 2008; Horne and Dabdub, 2017; McNeill, 2017; Shiraiwa et al., 2017). Numerous epidemiological studies have established consistent associations between PM and various health diseases, especially cardiorespiratory illnesses (Brunekreef, 2005; Franchini and Mannucci, 2009; Langrish et al., 2012; Ostro et al., 2011; Willers et al., 2013). Once inhaled, PM notably have the capacity to generate reactive oxygen species (ROS), which leads to pro-inflammatory responses that can ultimately result in apoptosis (Ayres et al., 2008; Jin et al., 2018; Nel, 2005; Piao et al., 2018; Yang et al., 2018). Investigating the PM oxidative potential (OP) in light of their major emission sources at various urban environments can then provide valuable information to instigate air pollution abatement policies limiting health outcomes. However, spatially-resolved PM source apportionment at a city-scale remains a challenging task (Dai et al., 2020b, 2020a; Pandolfi et al., 2020).

Receptor models demonstrated their ability to extract information by variable reduction techniques, especially in large datasets, in different branches of scientific research. In particular, the Positive Matrix Factorization (PMF) model is widely used in many studies to determine the contribution of emission sources in PM, based on the characterization of chemical tracers in a series of PM samples (Belis et al., 2014, 2020; Hopke, 2016; Pindado and Perez, 2011; Saeaw and Thepanondh, 2015; Weber et al., 2019). The option of refining source profiles by adding constraints have further improved the accuracy of identifying sources (Charron et al., 2019; Marmur et al., 2007; Weber et al., 2019; Zhu et al., 2018), especially when specific chemical species and unique tracers are included (Bullock et al., 2008). In fact, the PMF model has shown good strengths in both rural and urban environments (Pindado and Perez, 2011; Schauer and Cass, 2000), however, there are limited studies in cities at a fine-scale resolution that allows the assessment of local variabilities in a metropolitan area.

The city of Grenoble (France), with a complex topography and marked seasonal cycles of particulate pollution, offers interesting opportunities to explore the capability of PMF to resolve both the small spatial and large temporal scales of variabilities of the contribution of PM sources with the possibility of using additional tracers. Specific meteorological conditions, topography, and local sources impact the local PM chemistry in the atmosphere thereby requiring additional sources to properly scrutinize these local variations in urban environments. Further, previous works were already conducted in the area using extended PMF (Srivastava et al., 2018b; Weber et al., 2019), providing useful benchmark indicators.

The application of PMF requires to accurately consider a wide range of chemical components in PM, particularly for its organic fraction (Seinfeld and Pankow, 2003), consisting of complex mixtures especially in urban environments (Schauer and Cass, 2000; Zheng et al., 2004). In fact, around 80% of organic matter (OM) generally remains unidentified at the molecular level (Chevrier, 2016; Golly et al., 2019) resulting in misclassification or several un-apportioned sources of $PM_{10}$. Additionally, the difference in formation pathways of PM components may limit the identification of sources of PM, especially the secondary organic carbon (SOC) fraction, without the use of relevant organic tracers (Srivastava et al., 2018a; Wang et al., 2017a).





Different organic tracers have already been integrated in previous PMF studies, allowing to resolve specific sources of organic
aerosols that cannot be easily identified, such as primary biogenic aerosols and products of secondary processes in the
atmosphere (Waked et al., 2014; Belis et al., 2019; Golly et al., 2019; Hu et al., 2010; Weber et al., 2019).
In particular, Srivastava et al., 2018b was able to differentiate between different types of primary and secondary organic
fractions at a Grenoble urban background site, after analysing about 150 organic markers (and selecting 25 of them for the
final PMF run). Such studies are highly labour-intensive and often require the use of costly analytical devices and methods,
whereas some of the missing key molecular markers might still be obtained using simpler and/or more targeted techniques.
Moreover, the usefulness of these organic tracers in PMF analysis requires extensive methodological exploration, in terms of
their applicability as source tracers considering the much lower variability of their concentrations compared to other traditional
tracers.
In this paper, we present results of a study conducted over one year at three sites within 15 km of each other in the Grenoble
metropolitan area within the framework of the Mobil'Air project (available in https://mobilair.univ-grenoble-alpes.fr/, last
access: 02 November 2020). The sources of $PM_{10}$ were apportioned considering major chemical components contributing to
the PM mass, including organic and elemental carbon, ions, a condensed set of commonly-used organic markers (anhydride
monosaccharides, polyols, MSA), and metals. Additional fit-for-purpose tracers, including free cellulose and several organic
acids, were also added in the PMF input datasets to tackle specific sources that are difficult to discriminate using a traditional
PMF dataset only. Results obtained from this improved PMF analysis were then use to investigate the spatial and seasonal
variabilities in the source contributions for different urban typologies inside a metropolitan area. The overall outputs of this
study could be of interest to policy makers in providing vital information for designing effective particulate matter control
strategies including the setup of low emission zones and an opportunity to acquire more knowledge about the associations of
these emissions to other emerging health-based metrics (e.g., OP of PM) at a city scale as presented in the companion paper
(Borlaza et al., in prep).
**2 Methodology**
**2.1 $PM_{10}$ sample collection**
The metropolitan area of Grenoble, regarded as the capital of the French Alps, has a population of about 440,000 inhabitants.
The city itself presents a low altitude range (between 204 and 600 meters above sea level) but is located in an alpine
environment (Figure 1), surrounded by several mountain ranges, including Chartreuse (north), Vercors (south and west), and
Belledonne (east). These mountains restrict the movement of air heavily affecting the local meteorology and favouring the
development of atmospheric temperature inversions with entrapment of pollutants in the valley, particularly in the winter
(Bessagnet et al., 2020). During this study, a $PM_{10}$ sampling campaign was conducted in the Grenoble area at three sites
selected to represent various urban typologies, including: Les Frênes (LF, urban background site), Caserne de Bonne (CB,
urban hyper-center), and Vif (peri-urban area). These sites are all within a 15-km range from the city center. LF is a long-





standing reference urban background site for the regional air quality monitoring network (Atmo Auvergne Rhône-Alpes),
nearby a park at the outer fringe of the city. Vif is a peri-urban site, with suburban housings close to rural areas. However, this
site could potentially receive industrial emissions from a nearby chemical industrial area (<6 km) in the air flux within this
North – South valley. Substantial influence of biogenic emissions could also be expected as this site is in-between the foot of
Vercors and Belledone national parks. Lastly, while in a pedestrian area, the site of CB is in the hyper-center of Grenoble and
exposed to traffic emissions from the nearby boulevards.
The daily (24-h) $PM_{10}$ sampling collection was conducted from February 28, 2017 to March 10, 2018 (starting at 00:00 local
time) with an average 3-day sampling interval. A total of 125, 127 and 127 samples were collected during this year-long
campaign at LF, CB, and Vif, respectively. The $PM_{10}$ collection was performed using high volume samplers (Digitel DA80,
30 $m^3\,h^{-1}$) onto 150 mm-diameter pure quartz fibre filters (Tissu-quartz PALL QAT-UP 2500 diameter 150 mm). All filter
handling procedures of filters were strictly under quality control assurance procedures to avoid any possible contamination. In
particular, filters were preheated at 500 °C for 12 hours before use to avoid organic contamination. At least 20 field blank
filters were collected at each site to determine detection limits (DL) and to check for the absence of contamination during
sample transport, setup, and recovery. After particle collection, filter samples were wrapped in aluminium foil, sealed in zipper
plastic bags, and stored at <4 °C until further chemical analysis. Complementary measurements at the sampling sites notably
included the total $PM_{10}$ mass concentration measured using tapered element oscillating microbalances equipped with filter
dynamics measurement systems (TEOM-FDMS) (Grover, 2005).

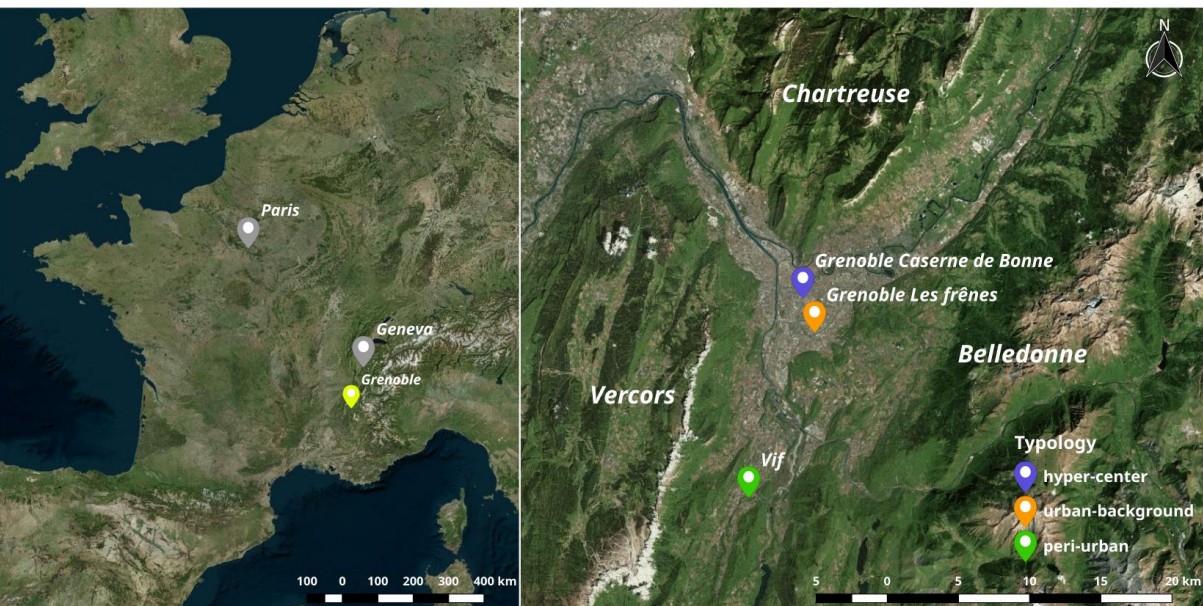


**Figure 1: Grenoble, the city where the sampling was made, placed on a European Map (left), and PM monitoring sites (right): Les**
**Frênes or LF (background), Caserne de Bonne or CB (hyper-center), and Vif (peri-urban). Image credit: Bing™ Aerial. © Microsoft**




## 2.2 Classical set of chemical analyses

Sampled filters were subjected to various chemical analyses for the quantification of the major chemical constituents and
specific chemical tracers of sources needed for PMF studies.
The carbonaceous fractions (organic carbon (OC) and elemental carbon (EC)) were analysed with a Sunset Lab analyser
(Aymoz et al., 2007; Birch and Cary, 1996) using the EUSAAR2 thermo-optical protocol (Cavalli et al., 2010). Total organic
matter (OM) in daily ambient aerosols were estimated by multiplying the OC mass by a fixed conversion factor of 1.8 based
on findings obtained from previous studies (Favez et al., 2010; Putaud et al., 2010).
A solid/liquid extraction was performed on 11.34 cm$^2$ punches soaked in a 10 ml of ultra-pure water under vortex agitation for
20 minutes. The extract was then filtered with a 0.25 µm porosity Acrodisc (Milipore Millex-EIMF) filter. The major ionic
components were measured by ion chromatography (IC) following a standard protocol described in Jaffrezo et al. (1998)and
Waked et al. (2014), using an ICS3000 dual channel chromatograph (Thermo-Fisher) with AS11HC column for the anions
and CS12 for the cations. This technique allowed the quantification of sodium (Na$^+$), ammonium (NH$_4^+$), potassium (K$^+$),
magnesium (Mg$^{2+}$), calcium (Ca$^{2+}$), chloride (Cl$^-$), nitrate (NO$_3^-$), sulfate (SO$_4^{2-}$), and methane sulfonic acid (MSA).
Furthermore, anhydro-sugars and saccharides were analysed by a High Performance Liquid Chromatography with Pulsed
Amperometric Detection (HPLC-PAD), using a Thermo-Fisher ICS 5000$^+$ HPLC equipped with 4 mm diameter Metrosep
Carb 2×150 mm column and 50 mm pre-column in isocratic mode with 15% of an eluent of sodium hydroxide (200 mM) and
sodium acetate (4 mM) and 85% water, at 1 ml min$^{-1}$. This method notably allowed the quantification of anhydrous saccharides
(levoglucosan and mannosan), polyols (arabitol and mannitol), and glucose as tracers of biomass burning and primary biogenic
aerosols (Samake et al., 2018; Waked et al., 2014).
Finally, major and trace elements were analysed after mineralization of a 38 mm diameter punch of each filter, using 5 ml of
HNO$_3$ (70%) and 1.25 ml of H$_2$O$_2$ during 30 minutes at 180 °C in a microwave oven (microwave MARS 6, CEM). The analysis
of 18 elements (Al, As, Ba, Cd, Cr, Cu, Fe, Mn, Mo, Ni, Pb, Rb, Sb, Se, Sn, Ti, V, and Zn) was performed on this extract using
inductively coupled plasma mass spectroscopy (ICP-MS) (ELAN 6100 DRC II PerkinElmer or NEXION PerkinElmer) in a
way similar to that described by Alleman et al. (2010).

## 2.3 Additional set of analyses of organic tracers

### 2.3.1 Organic acids

The analysis of a large array of organic acids (including pinic and phthalic acids, and 3-MBTCA) was conducted using the
same water extracts as for IC and HPLC-PAD analyses. In brief, this was performed by HPLC-MS (GP40 Dionex with a LCQ-
FLEET Thermos-Fisher ion trap), with negative mode electrospray ionization. The separation column is a Synergi 4 µm Fusion
– RP 80A (250×3 mm ID, 4 µm particle size, from Phenomenex). An elution gradient was optimized for the separation of the
compounds, with a binary solvent gradient consisting of 0.1% formic acid in acetonitrile (solvent A) and 0.1% aqueous formic


acid (solvent B) in various proportions during the 40-minute analytical run. Column temperature was maintained to 30 °C.
Eluent flow rate was 0.5 ml min$^{-1}$, and injection volume was 250 µl. Calibrations were performed for each analytical batch
with solutions of authentic standards. All standards and samples were spiked with internal standards (phthalic-3,4,5,6-d$^4$ acid
and succinic-2,2,3,3-d$^4$ acid). The calculation of the final atmospheric concentrations was corrected with the concentrations of
internal standards and of the procedural blanks, taking also into account the extraction efficiency varying between 76-116%
(depending on the acid).

### 2.3.2 Cellulose

The concentration of cellulose within PM$_{10}$ samples was quantified based on a protocol improving the procedure proposed by
Kunit and Puxbaum (1996). Cellulose was extracted from the filter in an aqueous solution, which was then processed in several
solutions of enzymes in order to break-down the cellulose into glucose units. Resulting glucose concentration was quantified
using an HPLC-PAD technique. To do so, a 21 mm diameter punch was first extracted for 40 minutes using an ultrasound bath
in 3 ml of an aqueous solution with thymol buffer (pH 4.8). Then two enzymes solutions (cellulase (Sigma Aldrich, C2730)
with 20 µl of an aqueous solution at 70 units g$^{-1}$) and glucosidase (Sigma Aldrich, 49291), with 60 µl of an aqueous solution
at 5 units g$^{-1}$) are added into the solution. The solution was then incubated at 50 °C for 24 hours for the hydrolysis to occur.
The hydrolysis is stopped by placing the solution in an oven at 100 °C for 45 minutes. The solution was then centrifuged (7000
rpm) for 15 minutes, and carefully extracted out using a syringe before being analysed with an HPLC-PAD instrument. The
procedural blanks are greatly improved when the enzymes stock solutions are filtered to lower their glucose content. This is
performed with a series of cleaning steps (n=10) by tangential ultrafiltration in a Vivaspin 15R tube at 7000 rpm in Milli-Q
water.
The HPLC-PAD (Dionex DX500) is equipped with a Methrom column (250 mm long, 4 mm diameter), with an isocratic run
of 40 minutes with the eluents A (50%, 18mM NaOH), B (25%, 100 mM NaOH + 150mM NaAc), and C (25%, 220 mM
NaOH). Column temperature is maintained at 30 °C. Eluent flow rate is 1 ml min$^{-1}$, and injection volume is 250 µl. Each
analytical batch also includes standard glucose solutions as well as standard cellulose solutions (using 20 µm beads, Sigma
Aldrich, S3504) that have been processed like the real samples in order to determine the specific efficiency of the cellulose-
to-glucose enzymatic conversion for each batch. The final calculation of the atmospheric concentration of the free cellulose
takes this conversion efficiency into account. It varied according to the batch, generally ranging from 65–80%. The calculation
of the cellulose concentration also takes into account the initial concentrations of atmospheric glucose of each sample,
determined in parallel with the HPLC-PAD analysis of sugars and polyols as described above. Finally, field and procedural
blanks are also taken into account.

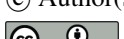



**2.4 Source apportionment**
**2.4.1 PMF input dataset**
Source apportionment of $PM_{10}$ was conducted using the United States Environmental Protection Agency (US-EPA) software
PMF 5.0 (Norris et al., 2014), aiming at the identification and quantification of the major sources of $PM_{10}$ for the three urban
sites in the Grenoble basin. Briefly, PMF is based on the factor analysis technique (Paatero and Tapper, 1994) applying a
weighted least-squares fit algorithm allowing the resolution of Eq. S1 (see supplementary information (SI)). In our study, 35
chemical species were used as input variables, namely OC*, EC, ions ($Na^+$, $K^+$, $NH_4^+$, $Mg^{2+}$, $Ca^{2+}$, $NO_3^-$, $SO_4^{2-}$ and $Cl^-$), trace
metals (Al, As, Cd, Cr, Cu, Fe, Mn, Mo, Ni, Pb, Rb, Sb, Se, Sn, Ti, V and Zn) and organic tracers (MSA, levoglucosan,
mannosan, polyols (sum of arabitol and mannitol), pinic acid, 3-MBTCA, phthalic acid, and cellulose), as summarized in Table
S1 in SI. We assumed that arabitol and mannitol originated from the same source, and hence combined them into one
component labelled as "polyols" (Samaké et al., 2019). In order to avoid double counting of carbon mass, OC* was calculated
using Eq. S2. The uncertainties of the input variables were calculated using Eq. S3 (Gianini et al., 2012). Finally, the species
displaying a signal-to-noise ratio (S/N) lower than 0.2 were discarded and those with S/N between 0.2 and 2 were classified
as "weak" variables (and then down-weighted applying 3-fold uncertainties).
**2.4.2 Set of constraints**
Since mixing issues between factors are inherent to PMF (i.e., collinearity due to meteorological conditions) and to possible
rotational ambiguity in the solution, we applied a set of constraints to the selected best base case solutions thanks to the ME-2
solver (Paatero, 1999). On top of the constraints defined in Weber et al. (2019), who applied a minimum set of constraints to
a large series of data sets within the SOURCE program, we added specific constraints for the traffic factor, derived from a
previous study in Grenoble dedicated to traffic emissions (Charron et al., 2019), as summarized in Table 1. These constraints
were applied similarly to the data sets from the 3 sites. This allows the orientation of the PMF solution towards more stable
and environmentally realistic profiles.

**Table 1: Summary of the applied chemical constraints on source-specific tracers in the PMF factor profiles.**

| Factor profile | Element | Type | Value |
|---|---|---|---|
| Biomass burning | Levoglucosan | Pull up maximally | (% dQ 0.50) |
| Biomass burning | Mannosan | Pull up maximally | (% dQ 0.50) |
| Primary biogenic | Levoglucosan | Set to zero | 0 |
| Primary biogenic | Mannosan | Set to zero | 0 |
| Primary biogenic | Polyols | Pull up maximally | (% dQ 0.50) |
| Primary biogenic | EC | Pull down maximally | (% dQ 0.50) |
| MSA-rich | MSA | Pull up maximally | (% dQ 0.50) |
| MSA-rich | Levoglucosan | Set to zero | 0 |
| MSA-rich | Mannosan | Set to zero | 0 |



| MSA-rich | Polyols | Pull down maximally | (% dQ 0.50) |
|---|---|---|---|
| MSA-rich | EC | Pull down maximally | (% dQ 0.50) |
| Nitrate-rich | Levoglucosan | Set to zero | 0 |
| Nitrate-rich | Mannosan | Set to zero | 0 |
| Mineral dust | Ti | Pull up maximally | (% dQ 0.50) |
| Primary traffic | Levoglucosan | Set to 0 | 0 |
| Primary traffic | Mannosan | Set to 0 | 0 |
| Primary traffic* | Cu | Pull up maximally | (% dQ 0.50) |
| Primary traffic* | Fe | Pull up maximally | (% dQ 0.50) |
| Primary traffic* | Sn | Pull up maximally | (% dQ 0.50) |
| Primary traffic* | $Ca^{2+}$ | Pull down maximally | (% dQ 0.50) |
| Primary traffic | Cu/Fe | Set to value | 0.046 (% dQ 0.50) |
| Primary traffic | Cu/Sn | Set to value | 5.6 (% dQ 0.50) |
| Primary traffic | Cu/Sb | Set to value | 12.6 (% dQ 0.50) |
| Primary traffic | Cu/Mn | Set to value | 5.7 (% dQ 0.50) |
| Primary traffic | OC*/EC | Set to value | 0.44 (% dQ 0.50) |

Note: *Only applied in Vif (peri-urban) site

### 2.4.3 Criteria for a valid solution

Solutions with a total number of factors between 7 and 12 were tested for the determination of the base cases. During factor selection, the $Q/Q_{exp}$ ratio (<1.5), the geochemical interpretation of the factors, the weighted residual distribution, and the total reconstructed mass were evaluated. Finally, the optimal solutions obtained for each urban site was subjected to error estimation to ensure stability and accuracy of the solutions, using displacement (DISP) and bootstrapping (BS) methods. The DISP analysis evaluates that no swapping had occurred in any of the factors. Solutions with >80 out of 100 BS mapped factors were considered appropriate solutions. The final retained optimal solutions after the application of constraints fulfilled the recommendations of the European guide on air pollution source apportionment with receptor models (Belis et al., 2014). The sensitivity of the solutions to the applied constraints was also carefully evaluated by comparison between the base and constrained cases. More information about the source apportionment methodology is provided in the SI.

### 2.4.4 Similarity assessment

A test of similarity between source profiles, based on their specific chemical relative mass composition at each site was performed by comparing the Pearson distance (PD) and standardized identity distance (SID) in order to evaluate the variability of the solutions across these different urban environments. The PD and SID were calculated using Eq. S4 (Belis et al., 2015). The PD metric represents the sensitivity of a chemical profile based on the differences in the major mass fractions of PM, whereas the SID represents the sensitivity to all components (hence taking into account trace species). Homogenous profiles that are stable over different site types are expected to have PD<0.4 and SID<1.0 (Pernigotti and Belis, 2018). Conversely, factors outside of this range are considered to have heterogeneous profiles.



**2.4.5 Estimation of the contribution uncertainties**

The BS profiles uncertainties for the obtained solutions are presented in the SI (S3), in the form of mean±std of the 100 BS for all sites. As PMF5.0 does not directly output this to the user, we provided an estimate of the contribution uncertainties based on the method presented in Weber et al. (2019). During the BS estimation, both the $G$ and $F$ matrices are available, however only the $F$ matrix is given back to the user (the $G$ matrix being used internally to map the different profiles). Hence, the daily contributions of each of the species are estimated using:

$$X_{BSi} = G_{ref} \times F_{BSi}$$

where $F_{BSi}$ is the profile of the bootstrap i, and $X_{BSi}$ is the time series of each species according the reference contribution $G_{ref}$ and the bootstrap run $F_{BSi}$. Similarly, the DISP contribution uncertainties are given by the reference contribution $G$ multiplied by the lower and upper limits of the DISP result for each species.

**3 Results and discussion**

**3.1 General evolution of concentrations of $PM_{10}$ and chemical species**

The daily $PM_{10}$ mass concentrations at the three measurement sites, determined with the TEOM-FDMS for the dates of filter sampling, ranged from 3-61 µg m$^{-3}$ with an overall average of 14±9 µg m$^{-3}$ during the sampling period. Average $PM_{10}$ levels were the highest at the urban hyper-center site (CB) (16±10 µg m$^{-3}$), followed by the urban background site (LF) (14±8 µg m$^{-3}$), and the peri-urban site (Vif) (13±9 µg m$^{-3}$). Annual averages of $PM_{10}$ mass concentrations and chemical compositions at all sites and at individual urban sites are shown in Table S2 in SI. The sites in this study showed minimal exceedances of the current $PM_{10}$ European limit value of 40 µg m$^{-3}$ (3.7%, 1.6%, and 1.6% of measurement days at the LF, CB, and Vif sites, respectively). Most of these exceedances occurred during the winter season indicating the necessity to additionally implement season-specific regulations for $PM_{10}$ emission reductions. Organic matter (OM) was the largest contributor in $PM_{10}$ and accounted for 54%, 51%, and 56% of mass concentration on an annual basis in LF, CB, and Vif, respectively. This contribution was followed by contributions from the major inorganic species ($NH_4^+$, $NO_3^-$, and $SO_4^{2-}$), suggesting strong influence from long-range transport of pollutants. An extensive description of the $PM_{10}$ chemistry in the Grenoble basin has already been presented in Srivastava et al. (2018b) for the years 2013–2014 at the LF site. Our results showed notable similarities for most chemical species for the year 2017–2018, especially in terms of seasonal variations and respective contribution of chemical species to $PM_{10}$ mass concentrations. Therefore, we will only describe these aspects briefly in this paper.

First, the time series analysis of $PM_{10}$ and its chemical composition in the Grenoble basin during the sampling period showed mild to strong seasonal trends. Part of it can be attributed to the atmospheric dynamics in the area given its alpine environment resulting in atmospheric temperature inversions that are especially common in winter. In the absence of strong winds during the winter season (especially during anti-cyclonic periods), higher concentrations of air pollutants could be expected. Indeed,

$PM_{10}$ concentrations were higher during the colder months (October to April) with an average of 17±10 μg m⁻³ and lower
during the warmer months (May to September) with an average of 10±4 μg m⁻³.

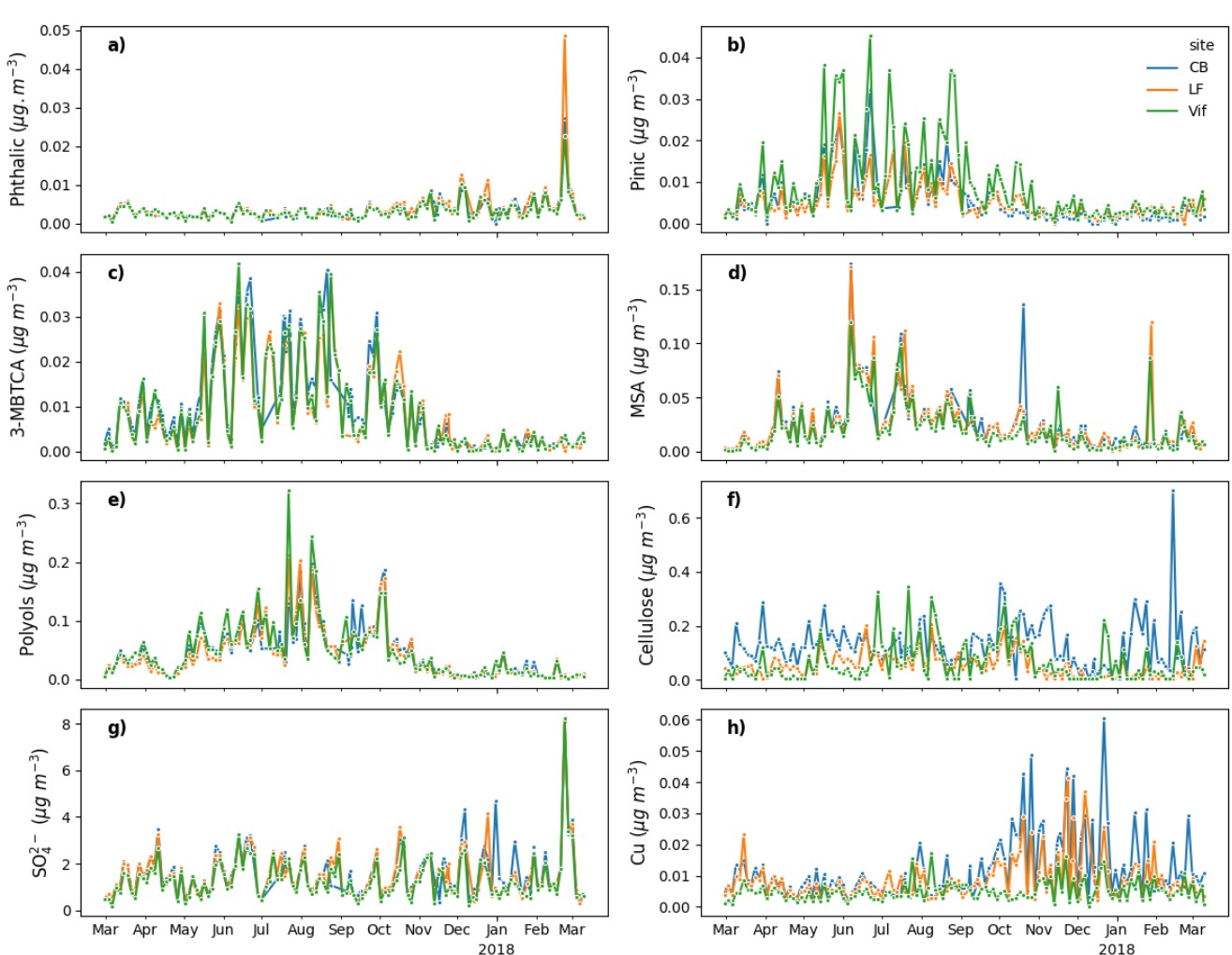


**Figure 2: Temporal evolutions of a) phthalic acid, b) pinic acid, c) 3-MBTCA, d) MSA, e) polyols (arabitol+mannitol), f) cellulose,**
**g) SO₄²⁻ and h) Cu in the three urban sites in the Grenoble basin (LF in orange, CB in blue, and Vif in green).**


We observed a strong seasonality for some chemical species with higher concentrations during the colder months including
OC*, EC, K⁺, NO₃⁻, NH₄⁺, levoglucosan, mannosan, and phthalic acid. These species are commonly associated with primary
emissions during the process of biomass burning (OC, EC, K⁺, levoglucosan, mannosan) and secondary atmospheric
processing (NO₃⁻, NH₄⁺, phthalic acid). Alternatively, specific species with higher concentrations during warmer months
include MSA, polyols, 3-MBTCA, and pinic acid. These species are known to be products of a wide range of photochemical





reactions in the atmosphere partly formed by OH-initiated oxidation (Atkinson and Arey, 1998; Szmigielski et al., 2007) and
can be explained by enhanced photochemical production due to an increase of temperature-dependent hydroxyl radical (OH)
concentration. A summary of temporal evolutions of the concentration for some species including $SO_4^{2-}$, Cu, cellulose, polyols,
3-MBTCA, pinic acid, and phthalic acid is shown in Figure 2.
Second, the Pearson correlation coefficients of the temporal evolution of each specie across sites is presented in Figure 3.
Similarity of temporal trends and strong correlations of $PM_{10}$ components between our 3 sites indicates the influence of large
scale transport processes or possible uniform distribution of some emission sources in the Grenoble area. Further, the
accumulation and removal processes of the PM may be driven by similar season-specific environmental conditions at a local
scale. A strong correlation was observed in OC*, EC, ions, polyols, levoglucosan, mannosan, 3-MBTCA, phthalic acid, and
pinic acid between sites suggesting similar origins and atmospheric processes affecting the concentrations of these species.
The three sites seem to be equally impacted by long range transport since concentration of $SO_4^{2-}$ appears almost identical. We
also clearly see relatively similar temporal trends for the organic acids (MSA, pinic, and 3-MBTCA). Notably, we also
observed an important episode in phthalic acid in late February 2018 affecting all the three sites. Conversely, cellulose and
most metal species showed weak to mild correlations between sites, possibly indicating that the sources of these species are
highly localized, with a potential impact that is variable at a city-scale. Particularly, cellulose presents similar order of
magnitude at the three sites but presents higher concentration at CB, especially during winter. A few metals only showed
strong correlations between LF and CB, but not with Vif, such as Al, Cu, Fe, Rb, and Sb which are tracers of road transport
activity or biomass burning emissions. Specifically, Cu concentrations are similar at the three sites during summer, but presents
significantly lower concentration in Vif compared to the two urban sites of CB and LF during winter.

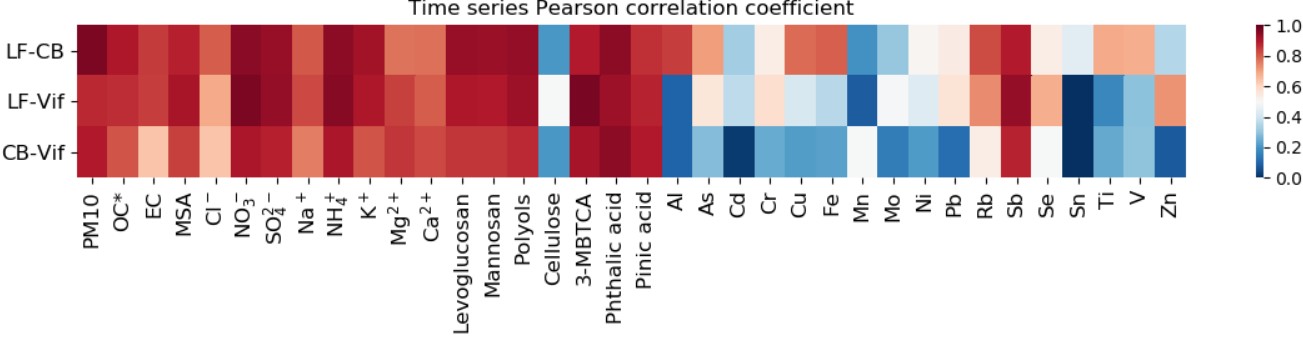


**Figure 3: Heat map of the time series Pearson correlation coefficient of $PM_{10}$ and its chemical composition between LF and CB (LF-**
**CB), LF and Vif (LF-Vif), and CB and Vif (CB-Vif).**




**3.2 PM₁₀ source apportionment**

In the following sections, a description of the best PMF solution obtained after application of constrains is provided for each
of the 3 sites, together with a discussion about the factors that are associated with the added organic tracers (MSA, polyols,
cellulose, pinic and 3-MBTCA acids). The presentation of error estimations, chemical profiles, and temporal evolutions of the
PMF-resolved sources, and the discussion about the more classical factors can be found in the SI (S3).

**3.2.1 General description of the solutions**

The PMF model was applied independently on the data set of each three sites, using 35 chemical atmospheric compounds in
each site. The constrained solutions for each site consist of 11 factors, including common factors such as primary traffic,
biomass burning, nitrate-rich, sulfate-rich, aged sea salt, sea/road salt, and mineral dust. Also, with the use of biogenic tracer
species, we identified a primary biogenic factor and a MSA-rich factor, similar to the ones determined in Weber et al. (2019)
for each of 15 sites in France. We also determined a metals-rich factor, identified as an industrial factor, accounting for a very
small part of the PM₁₀ mass. Finally, using new organic proxies (pinic and 3-MBTCA acids), we identified a secondary
biogenic oxidation factor that is rarely described in other PMF studies. Table 2 shows a synthesis of the tracers used to identify
these 11 PMF-resolved factors that are found at each of the 3 sites.
Other solutions with fewer or greater number of factors were also investigated but these solutions were less defined, and factor
merging was often observed. The reconstructed PM₁₀ contributions from all sources with measured PM₁₀ concentration showed
very good mass closure in all sites (LF: r=0.99, n=125, p<0.05; CB: r=0.99, n=126, p<0.05; and Vif: r=0.99, n=126, p<0.05)
indicating very good model results.
This result is in line with a previous study in the city of Grenoble (Srivastava et al., 2018b), but with slight improvements in
the PM₁₀ mass closure (from r=0.93 to r=0.99). A complete comparison of the PMF-resolved sources between the two studies
is presented and discussed in SI (S4). The two sets of results are in good agreement, despite the samples being collected 4
years apart. There were several identified sources that are similar in both studies such as biomass burning, primary traffic,
mineral dust, aged sea salt, sulfate- and nitrate-rich (identified collectively as secondary inorganics in Srivastava et al. (2018b)),
and primary biogenic (identified as fungal spores and plant debris in Srivastava et al. (2018b)). Additionally, due to a number
of differences in the input variables used, there are some sources that are completely unique to each study. In particular, the
sources that we have uniquely identified are industrial, sea/road salt, MSA-rich, and secondary biogenic oxidation sources.
Conversely, Srivastava et al. (2018b) have uniquely identified two SOA sources: biogenic SOA and anthropogenic SOA. It
can be argued that the secondary biogenic oxidation source (11%) in our study and the biogenic SOA (12%) in Srivastava et
al. (2018b) are in some way similar, although different tracers were used to identify them. Particularly, Srivastava et al. (2018b)
identified the biogenic SOA source with high contributions from α-methylglyceric acid (α-MGA and 2-methylerythritol (2-
MT), hydroxyglutaric acid (3-HGA), while our study identified the secondary biogenic oxidation source with high





contributions from 3-MBTCA and pinic acid. While not uniquely identified in our study, the contributions of phthalic acid in
several common anthropogenic-derived sources (sulfate- and nitrate-rich) can also mark the potential contributions from
anthropogenic SOA sources. Finally, the considerable economic advantage in the specific organic tracers used in our study, in
terms of the type of chemical analyses performed, could assist future studies utilizing organic species in PMF.
It is also important to note that, although still in the acceptable range, the sulfate-rich factor obtained in our PMF results yielded
the most BS unmapped factors amongst the PMF-resolved factors (up to 25% for the CB site). This may be the sign of possible
mixing of different processes / sources in this factor.

**Table 2: Summary of PMF-resolved sources and their specific tracers.**

| Identified factors | Specific tracers |
|---|---|
| Biomass burning | Levoglucosan, mannosan, $K^+$, Rb, $Cl^-$ |
| Primary traffic | EC, $Ca^{2+}$, Cu, Fe, Sb, Sn |
| Nitrate-rich | $NO_3^-$, $NH_4^+$ |
| Sulfate-rich | $SO_4^{2-}$, $NH_4^+$, Se |
| Mineral dust | $Ca^{2+}$*, Al, Ti, V |
| Sea/road salt | $Na^+$, $Cl^-$ |
| Aged sea salt | $Na^+$, $Mg^{2+}$ |
| Industrial | As, Cd, Cr, Mn, Mo, Ni, Pb, Zn |
| Primary biogenic | Polyols, cellulose |
| MSA-rich | MSA |
| Secondary biogenic oxidation | 3-MBTCA, pinic acid |

Note: *Vif site did not have high loadings of $Ca^{2+}$ specie in this factor
**3.2.2 PM$_{10}$ contribution**
Biomass burning (17-26%), sulfate-rich (16-18%), and nitrate-rich (14-17%) sources were the highest contributors to the total
PM$_{10}$ mass on a yearly average in the Grenoble basin. Primary traffic (12-14%) and secondary biogenic oxidation (8-11%)
sources also contributed a relevant amount. Figure 4 presents a comparison of the source contributions in each site based on
mass concentration (in μg m$^{-3}$). These results are in line with recent studies leading to anthropogenic and SOA sources heavily
influencing urban air pollution in western Europe (Daellenbach et al., 2019; Golly et al., 2019; Pandolfi et al., 2020; Srivastava
et al., 2018b; Weber et al., 2019). The most notable difference across all sites is the sharp decrease of mineral dust in Vif
compared to the other two urban sites, and this is discussed further in section 3.4.1.





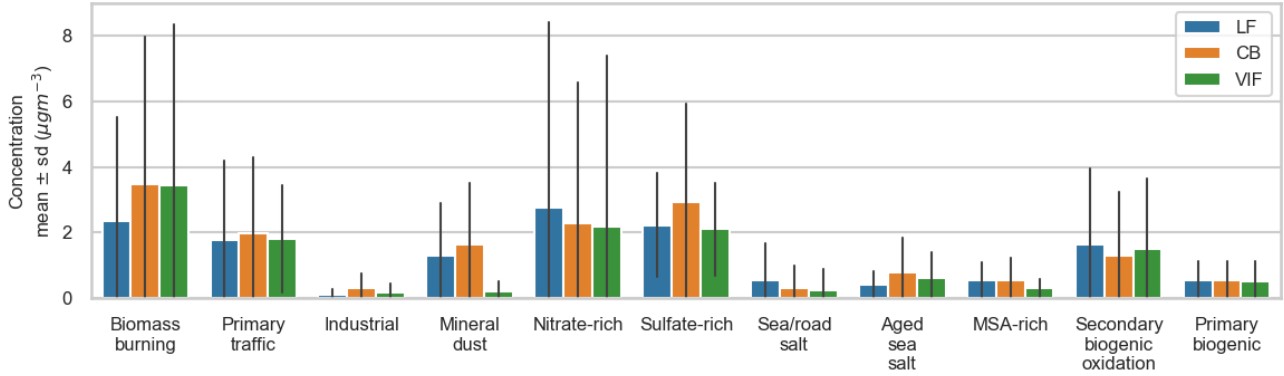

**Figure 4: Factor contributions in µg m⁻³ for the three sites (LF: blue, CB: orange, Vif: green). Bar plots depict the mean annual value and the standard deviation of daily variations.**

### 3.2.3 MSA-rich

This factor is identified with a high loading of MSA, a known product of oxidation of dimethylsulfide (DMS), commonly described as resulting from marine phytoplankton emissions (Chen et al., 2018; Li et al., 1993). Other chemical species with significant concentrations in this factor include sulfate and ammonium. Although a very useful tracer of marine biogenic sources, MSA showed in our series only weak to mild correlations with ionic species from marine aerosols such as $Na^+$ (r: 0.2–0.3) and $Mg^{2+}$ (r: 0.3–0.4). This suggests potential emissions originating from terrestrial biogenic sources instead, which has been similarly suggested before (Bozzetti et al., 2017; Golly et al., 2019), and/or from forest biota (Jardine et al., 2015; Miyazaki et al., 2012). On an annual scale, this factor accounted for 2-4% of the total mass of $PM_{10}$ and shows a strong seasonality with highest contributions during summer, reaching up to 53%, 57%, 52% of the total $PM_{10}$ mass in some specific days in LF, CB, and Vif, respectively. The similarity in the temporal distribution across sites, as shown in Figure S3.8, especially the summer peaks, could be linked to the influence of long-range transport of pollutants in the MSA-rich factor.

### 3.2.4 Primary biogenic

The primary biogenic factor was identified with high loadings of both polyols and cellulose (see Figure 5). Polyols (represented by the sum of arabitol and mannitol) are known as tracers of primary biological aerosols from fungal spores and microbes (Bauer et al., 2002; Igarashi et al., 2019). Polyols has been used in several studies as a tracer of biogenic sources, contributing in France within a range of 5-9% of $PM_{10}$ on a yearly average (Samaké et al., 2019, 2019; Srivastava et al., 2018b; Waked et al., 2014; Weber et al., 2019). Cellulose is a potential macro-tracer for plant debris from leaf litter and seed production (Kunit and Puxbaum, 1996; Puxbaum, 2003) that is very rarely used in source apportionment studies as of today, while it can represent





a large fraction of the PM mass in the coarse mode (Bozzetti et al., 2016), for example, it represents up to 6% during the warm
season in the Vif site.

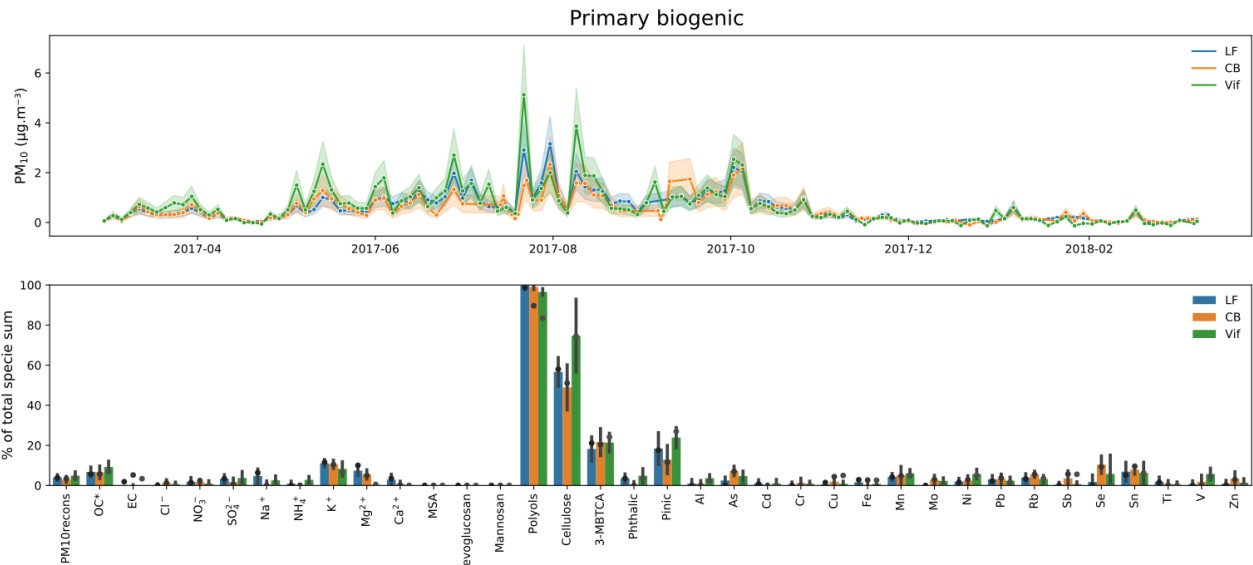

**Figure 5: Primary biogenic factor for the 3 urban sites. Top: Contribution to PM$_{10}$ given the mean and standard deviation of the**
**100 BS. Bottom: Percentage (%) of each specie apportioned by this factor (dots refer to the constrained run, bar plots refer to the**
**mean and error bars refer to the standard deviation of the 100 BS).**

A strong correlation was found in the temporal evolution of polyols across the 3 sites in our study indicative of large scale
impact of sources for these species (Samaké et al., 2019a,b). Conversely, cellulose concentrations present only weak
correlations across the 3 sites, possibly indicating that the influence of the sources of this specie might be more local. Although
polyols and cellulose are both tracers of biogenic sources, only a rather mild correlation (r=0.5) was found between these two
tracers, with seasonality of their concentrations being slightly different (Figure 2). It shows that the processes and the sources
are probably distinct for the two sets of chemical species. However, the PMF is not able to separate them, and this factor
includes most of the cellulose (58, 51, and 74 % in LF, CB, and Vif, respectively), and also most of the polyols (99, 90, and
83 % in LF, CB, and Vif, respectively). The remaining fraction of cellulose concentrations was included in the mineral dust
factor in LF and CB, and in the primary traffic factor in Vif, suggesting the possibility of resuspension processes for this
compound (see the SI for details). We can also note that the cellulose was not apportioned in the biomass burning factor, an
indication that it may not be emitted by this source.
Despite their slightly different origins, the PMF analysis captures the combined contribution of polyols and cellulose to a factor
that can be termed "primary biogenic sources". In this study, this factor accounted for 3-4% of the total mass of PM$_{10}$ on an
annual scale, and a strong seasonality was observed, with up to 18% (in LF), 8% (in CB), and 17% (in Vif) of the total PM$_{10}$


mass on average in summer, with specific days reaching up to 60% of PM$_{10}$ for example at the Vif site (see Figure 5). These
temporal variations are consistent with higher biological activity (increased production of fungal and fern spores, and pollen
grains) in this season due to increase in temperature and humidity (Graham et al., 2003; Verma et al., 2018). This may also be
attributed to an increased plant metabolic activity (production of plant debris from decomposition of leaves) and the proximity
to forested and agricultural areas of the sampling sites (Gelencsér et al., 2007; Puxbaum, 2003). Finally, one can note that the
chemical profiles also include some fractions of the tracers from secondary biogenic production (3-MBTCA and pinic acid),
indicative of some degree of mixing between primary and secondary biogenics.

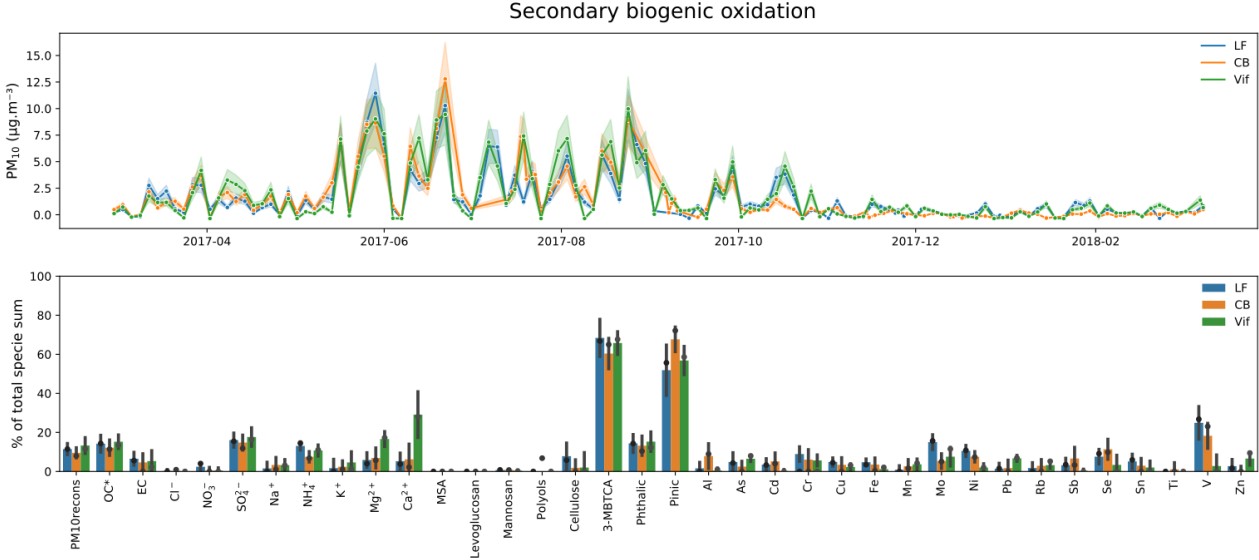

**Figure 6: Secondary biogenic oxidation factor for the 3 urban sites. Top: Contribution to PM$_{10}$ given the mean and standard deviation of the 100 BS. Bottom: Percentage (%) of each specie apportioned by this factor (dots refer to the constrained run, bar plots refer to the mean and error bars refer to the standard deviation of the 100 BS).**

### 3.2.5 Secondary biogenic oxidation

The secondary biogenic oxidation factor was identified with high loadings of 3-MBTCA and pinic acids (see Figure 6). Both
tracers of this factor are known to be products of secondary oxidation processes of alpha-pinene from various biogenic origins.
The apportionment of such a factor is not commonly achieved in receptor modelling using off-line tracers (van Drooge and
Grimalt, 2015; Heo et al., 2013; Hu et al., 2010; Srivastava et al., 2018a). On an annual scale, this factor accounted for 8-11%
of the total mass of PM$_{10}$, but can be as high as 58% (11 µg m$^{-3}$) on specific days (see Figure 6, Top). The strong correlation
between 3-MBTCA and pinic acids suggests similarity of origin of the secondary biogenic oxidation factor in the Grenoble
area, despite inter-site correlations for 3-MBTCA (older oxidation state of alpha-pinene, hence more homogeneous at the city





scale) being larger than that for pinic acid (former oxidation product, less homogeneous). Although significant portions (56-
72%) of these species (3-MBTCA and pinic acids) are in this secondary biogenic oxidation factor, there are still relevant
contributions in other factors, including primary biogenic, sulfate- and nitrate-rich, aged sea salt, and MSA-rich. Conversely,
the presence of phthalic acid contribution in this factor (around 10% of its concentration), which could be emitted directly
from biomass burning or formed during secondary processing from anthropogenic emissions (Hyder et al., 2012; Kleindienst
et al., 2007; Wang et al., 2017b; Yang et al., 2016), also suggests that the secondary biogenic oxidation factor may be affected
by these emissions. All of these indicate that the PMF process did not deliver a pure secondary biogenic oxidation factor, either
due to data processing limitation or because of real mixing of these sources in the PM.

**3.3 Re-assignment of factors thanks to the new proxies**
**3.3.1 Importance of the new proxy for factor identification**
With the use of these additional organic tracers, there are several added information drawn from the results of the PMF model.
First, the notable contributions of phthalic acid in several sources could further confirm the mixing influence of anthropogenic
processes in various sources of $PM_{10}$ such as sulfate- and nitrate-rich, but also with secondary biogenic oxidation sources.
Second, adding 3-MBTCA and pinic acids in the input variables allowed the identification of a significant secondary biogenic
oxidation factor that is generally difficult to identify with PMF studies of off-line samples. Comparisons already started with
the factors obtained by AMS studies (Vlachou et al., 2018), but more work remain to be done in order to evaluate their proper
correspondence.
**3.3.2 Comparison with a "classic" PMF solution**
In order to quantify the added value and the changes brought in by the additional tracers, a reference PMF using a chemical
data set (not including cellulose, pinic acid, phthalic acid, and 3-MBTCA) and parameters similar to that in the SOURCES
project (Weber et al., 2019) was performed, hereafter called "classic", and the results were compared with those from the
present study (called "orga"). Figure 7 shows the comparison of the yearly average mass contribution of the different factors
for these two approaches. A detailed comparison of chemical profiles between the "classic" and "orga" PMF runs in each site
is summarized in the SI (S3). One can see that most observations below are consistent in all three sites.
Some factors remain unaffected or only marginally modified: it is the case for the biomass burning source with a percentage
increase in contribution, only ranging from 1-14%, in the "orga" compared to the "classic" PMF run across all sites. The
primary biogenic source also posed an interesting case with a minimal decrease in contribution at 0.1 and 6% in the LF and
CB sites, respectively. However, adding more specific biogenic tracers changed the contribution of the primary biogenic factor
in Vif, from 1.1 µg m$^{-3}$ for the "classic" PMF down to 0.50 µg m$^{-3}$ for the "orga" PMF run, a value that is much more in line
with the contributions observed at the other sites (0.56 and 0.55 µg m$^{-3}$ in CB and LF, respectively). This further highlights




the usefulness of the additional organic tracers (e.g., addition of cellulose in the primary biogenic factor), especially for specific
site typologies.
Conversely, the most impacted factor is the sulfate-rich one, to a similar extent for the 3 sites with much higher mass fraction
in the "classic" PMF run in large part due to higher loadings of OC*. It may indicate possible merging with organic aerosol
sources in the "classic" PMF, as presented in a comparison of chemical profiles between the "classic" and "orga" PMF runs
in each site summarized in the SI (S3). Figure 7 shows that the differences are really close to the content of the new secondary
biogenic oxidation factor. Secondary aerosols, such as the sulfate-rich factor, can be transported over long distances and can
remain in the atmosphere for about a week (Warneck, 2000), allowing them to interact with numerous other species and
undergo different atmospheric oxidation processes. In fact, several studies have investigated various oxidation pathways of
sulfate-rich sources (Barker et al., 2019; Ishizuka et al., 2000; Schneider et al., 2001; Ullerstam et al., 2002, 2003; Usher et al.,
2002). In the SPECIEUROPE database, several studies have reported sulfate-rich sources influenced by a variety of different
fuel combustion sources (Bove et al., 2014; Pernigotti et al., 2016; Pey et al., 2013). It is, therefore, not surprising that part of
the matter in the sulfate-rich source was re-assigned to different other sources upon addition of the organic tracers in the "orga"
PMF run. A comparable study in Metz (France) also used another organic tracer (oxalate) to apportion a secondary organic
aerosol (SOA) source from PM, ascribing it possibly to both biogenic and anthropogenic emissions (Petit et al., 2019).

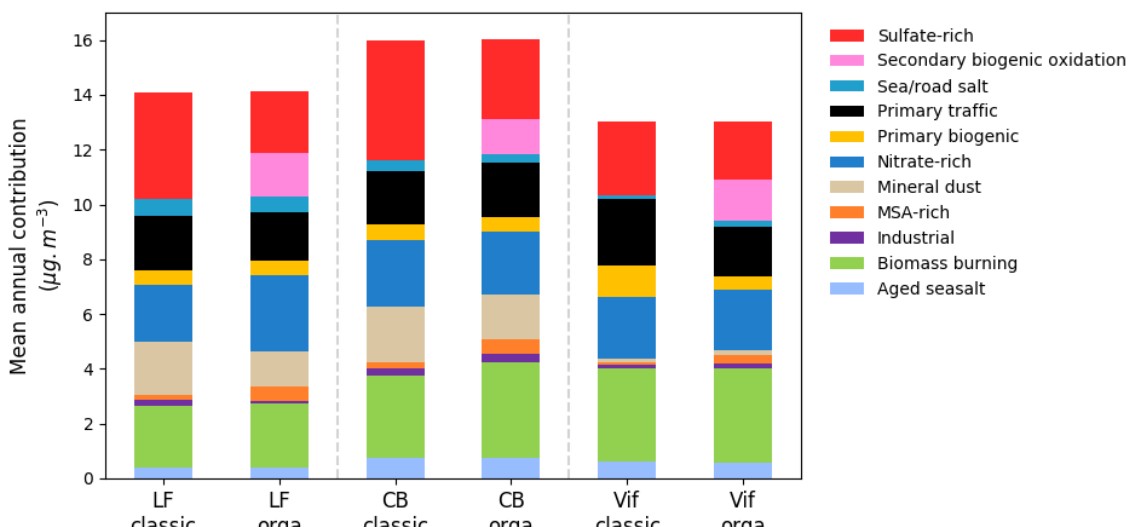

**Figure 7: Mean annual contribution (μg m⁻³) of PMF-resolved factors of PM$_{10}$ in the Grenoble basin using a classic set of input**
**variables similar to SOURCES ("classic") and using additional new organic tracers ("orga").**

We also observed an increase in the contributions of the MSA-rich factor at the three sites, with an increase in contributions
from specific inorganic species, such as $SO_4^{2-}$ and $NH_4^+$ (see Figure S3.8.1 in the SI). Conversely, a decrease in contribution





from polyols was observed in the chemical profile of primary biogenic factor in Vif (see Figure S3.7.1 in the SI). Results show that in the "classic" PMF run, the contribution of polyols was almost completely assigned to the primary biogenic factor (>94% of its total mass) while the "orga" PMF run resulted in a contribution of polyols to the MSA-rich factor of about 10% of its total mass.

Finally, there is also an observed re-assignment of the $Ca^{2+}$ specie that further refined specific factors in Vif. The mineral dust factor is often identified with high loadings of $Ca^{2+}$, however this is not the case for Vif, particularly for the "classic" PMF run (less than 1% of total $Ca^{2+}$, although attached with important uncertainties). With the addition of the organic tracers, there was an observed increase in the contribution of $Ca^{2+}$ in the mineral dust factor in Vif (see Figure S3.11.1), resulting to more than 20% of the total $Ca^{2+}$ apportioned in this factor (a value is still attached with important uncertainties). Interestingly, the contribution of $Ca^{2+}$ is mainly transferred from the primary traffic factor to the mineral dust factor, resulting in decreased contribution of the primary traffic factor in Vif from 2.4 µg m$^{-3}$ for the "classic" PMF down to 1.8 µg m$^{-3}$ for the "orga" PMF run, again to a value closer to the contributions at the other sites (2.0 and 1.8 µg m$^{-3}$ in CB and LF, respectively) (see Figure S3.2.1 in the SI).

### 3.3.3 Decrease of uncertainties

Another advantage of adding specific proxies in the PMF is the lowering of uncertainties associated with some other chemical species in some factors. Indeed, we observed a decrease of the BS uncertainties, notably for the OC* and also for some main tracers of sources in several profiles (see in the SI (S3)). The sulfate-rich is the most impacted factor when adding the new organic tracers and the higher uncertainties in the "classic" PMF run provided insights that this profile may have some internal mixing. Splitting this factor, thanks to the new organics, refined the sulfate-rich factor and strengthened the BS stability of this factor, decreasing the BS uncertainties.

Concerning the DISP, the range of uncertainties was also narrowed for 74% of the species in factors when comparing the "classic" and "orga" PMF. This decrease of uncertainties for the DISP when adding new variables was already observed by (Emami and Hopke, 2017), but our study additionally observe this in the BS error estimation. Overall, on top of being able to identify new factors, the addition of the new specific proxies in the PMF strengthened the confidence we have for all other factors.

### 3.4 Fine scale variability of the temporal contribution

Figure 3 indicates correlations of the concentrations for many chemical species across the sites. Additionally, the temporal evolution of the contribution of commonly resolved factors are further investigated in this section. Figure 8 presents the Pearson correlation coefficient of the contributions of the sources for the three pairs of sites. The sources that resulted to consistent strong correlations ($r > 0.77$) across all sites are biomass burning, nitrate-rich, aged sea salt, MSA-rich, secondary biogenic oxidation, and primary biogenic sources. The sea/road salt factor showed good correlations across the sites with a correlation coefficient ranging from 0.58 to 0.76.



487

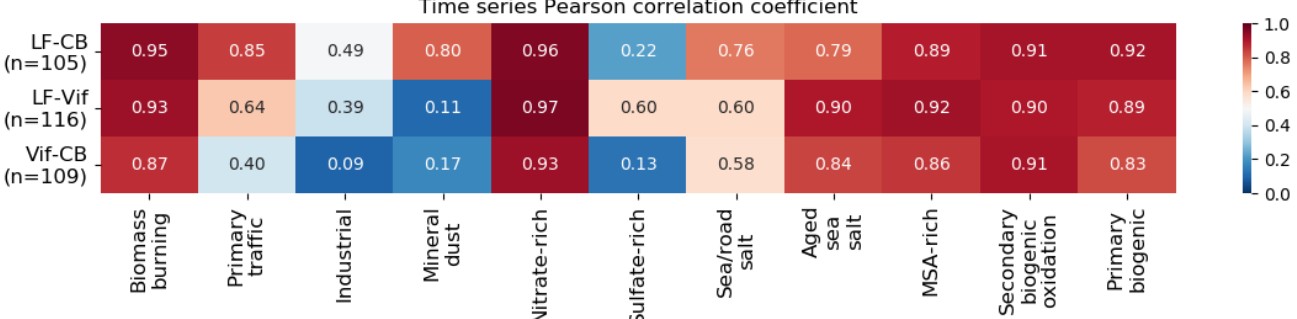

488

**Figure 8: Heat map of the time series Pearson correlation coefficient of all factor contributions between LF and CB (LF-CB), LF and Vif (LF-Vif), and CB and Vif (CB-Vif).**

491

Factors with strong seasonality appeared to be highly correlated between sites (biomass burning, nitrate-rich, MSA-rich, and primary biogenic). This tends to affirm that such factors are dominated either by large scale transport (i.e., nitrate-rich) or by a strong climatic determinant. It is interesting to note that the primary biogenic factor presents systematically a slightly lower correlation than the polyols (LF-CB: $r_{polyols}$=0.94 to $r_{primary\ biogenic}$=0.91, LF-Vif: $r_{polyols}$=0.92 to $r_{primary\ biogenic}$=0.88 and Vif-CB: $r_{polyols}$=0.87 to $r_{primary\ biogenic}$=0.82). This may suggest a secondary process or a combination of several different primary processes in the primary biogenic factor affecting the sites at different rates (Petit et al., 2019; Samaké et al., 2019). We also clearly see a stronger similarity between the two urban sites (LF and CB) compared to the peri urban one, notably for the primary traffic, mineral dust, and, to a lower extent, the industrial factor. This may be explained not only by the proximity of the location of the two former sites within the city, but also by their similarity in typology compared to the peri-urban site type in Vif. However, there are two factors that do not present good correlation between all sites.

One of them is the sulfate-rich factor which presents a similar contribution when comparing LF and Vif, but low-to-none correlation when compared to CB. A deeper analysis shows that the sulfate-rich, together with the nitrate-rich factor in CB, explains a large part of the winter spike of secondary inorganics (23/02/2018 to 24/02/2018), whereas in LF and Vif only the nitrate-rich factor explains most of it. This spike drives the Pearson correlation coefficient to a low value and without it, the correlation increases drastically (see Figure S5.1 in the SI for the full scatterplot). Some PMF solutions of the BS in LF and Vif also had this behaviour, but weren't chosen as the "best" solution. We propose two hypotheses for this difference: 1) during winter, some heterogeneous chemistry may take place in fog episodes in the Grenoble basin (resulting to episodic spikes in the $SO_4^{2-}$ contribution), that may not be spatially homogeneous at a city scale, leading to mixing of secondary sources, and 2) we have reached the limit of the PMF methodology to de-convolute further the secondary inorganics. Both hypotheses may be concurrent.





We note however that apart from these spikes, the $SO_4^{2-}$ apportioned by this factor at 3 sites are in good agreement, and are
within the uncertainties of each other (see Figure 9). This figure also highlights that the uncertainty for the $SO_4^{2-}$ in this factor
is higher for the CB site, as also shown in the chemical profile in Figure S3.6 in the SI.

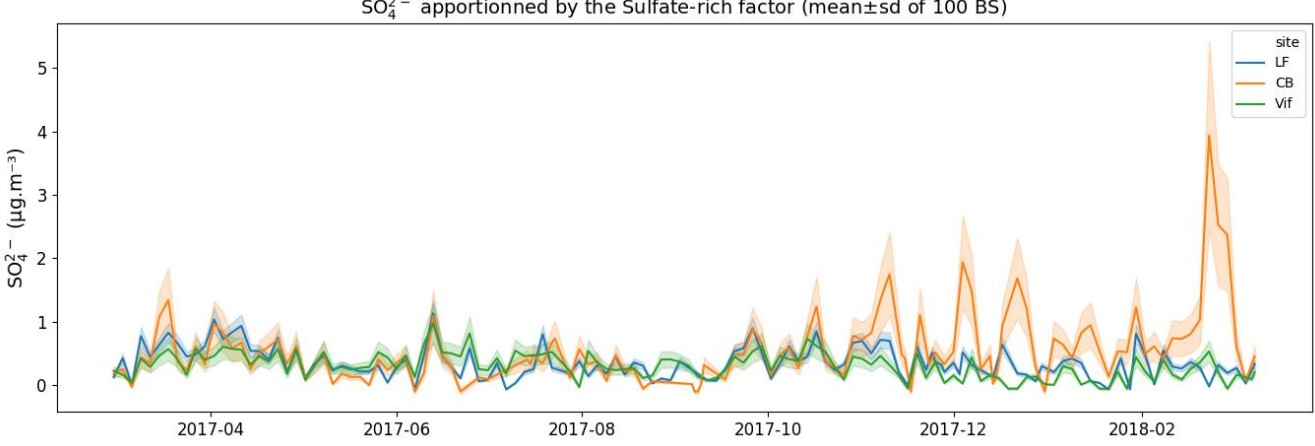


**Figure 9: $SO_4^{2-}$ apportioned by the sulfate-rich factor at the 3 sites, according to the uncertainties given by the 100 BS as shown by**
**the mean (solid line) and the standard deviation (shaded area).**

The second factor which showed low correlations between pairs of sites is mineral dust, specifically when comparing Vif to
the two other sites. This is in line with the difference in the $PM_{10}$ apportioned by this factor as shown in the previous section.
However, a closer look on the contribution scatterplot of $Vif_{Mineral\ dust}$ vs $CB_{Mineral\ dust}$ (see Figure S5.2 in the SI) highlights that
some events are very close to the 1:1 line. This is indicative of two regimes for mineral dust, with cases when the sources for
the urban and peri urban are being similar and cases when they are different. To investigate it further, a potential source
contribution function (PSCF) analysis of the mineral dust factor for the Vif and CB sites was performed in order to assess the
origin of air masses of this factor. It is presented in Figure 10. We can clearly see that for the Vif site, the main origin is Spain,
whereas the origin for CB is not well-defined. These PSCF pattern tends to indicate that the sources of the mineral dust factor
present a strong local component for the urban sites (CB and LF being very similar), while the origin of the mineral dust factor
in Vif appears to be mainly affected by long-range transport of dust only.





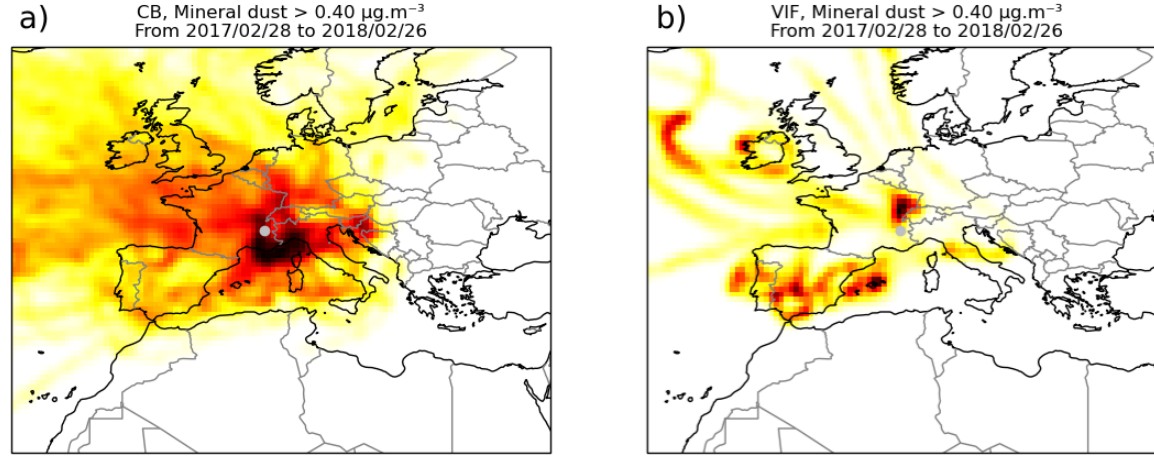

**Figure 10: The PSCF analysis of the days with a mineral dust loading higher than 0.4 μg m⁻³ for the CB site (a) and Vif (b). Darker shades indicate higher probability density of source origin.**

**3.5 Fine-scale variability of chemical profiles**

An additional similarity test was also performed to investigate the fine-scale variabilities of the chemical profiles of the factors. A similarity analysis at a regional scale in France identified stable chemical profiles obtained by PMF studies across many sites, corresponding to biomass burning, sulfate-rich, nitrate-rich, and fresh sea salt factors (Weber et al., 2019). In our study, a parallel analysis was performed in order to evaluate the stability of the chemical profiles of the identified factors in high proximity receptor locations. Briefly, PMF-resolved sources were compared for each pair of sites using both Pearson distance (PD) and standardized identity distance (SID) to obtain a similarity metric (PD-SID).

**3.5.1 (Dis-)similarity of the chemical profiles at the three sites**

Figure 11 presents the similarity plot (PD-SID) obtained for the 11 factors found in this study. The biomass burning factor yielded the most stable chemical profile in all the sites in the Grenoble basin, which is consistent. Other stable factors include sulfate- and nitrate-rich, primary biogenic, MSA-rich, and secondary biogenic oxidation. The industrial and sea/road salt factors, both marginally above the accepted PD metric, could be considered as having heterogeneous profiles based on the contributions of these sources to the total $PM_{10}$ in each site.

However, a clear heterogeneous chemical profile was found in the mineral dust, this further emphasized the difference in origin of this factor as previously discussed in section 3.4. More details about of the chemical profile of this factor can be found Figure S3.11 in the SI. One of the main differences is the lack of OC* in the Vif site compared to LF and CB sites, together with a much lower $Ca^{2+}$ contribution. Additionally, there is a lower $SO_4^{2-}$ apportioned in the mineral dust factor in Vif. The





only similarity between all the sites are the high loadings of Al, Ti and V, as well as important contributions from other crustal
metals (Fe, Ni, Mn). It also has to be noted that the cellulose is present up to about 20% of its total mass in the mineral dust
profile in the CB site, however the BS estimates indicate very important uncertainties for this specie in this factor (see Figure
S3.11 in the SI).

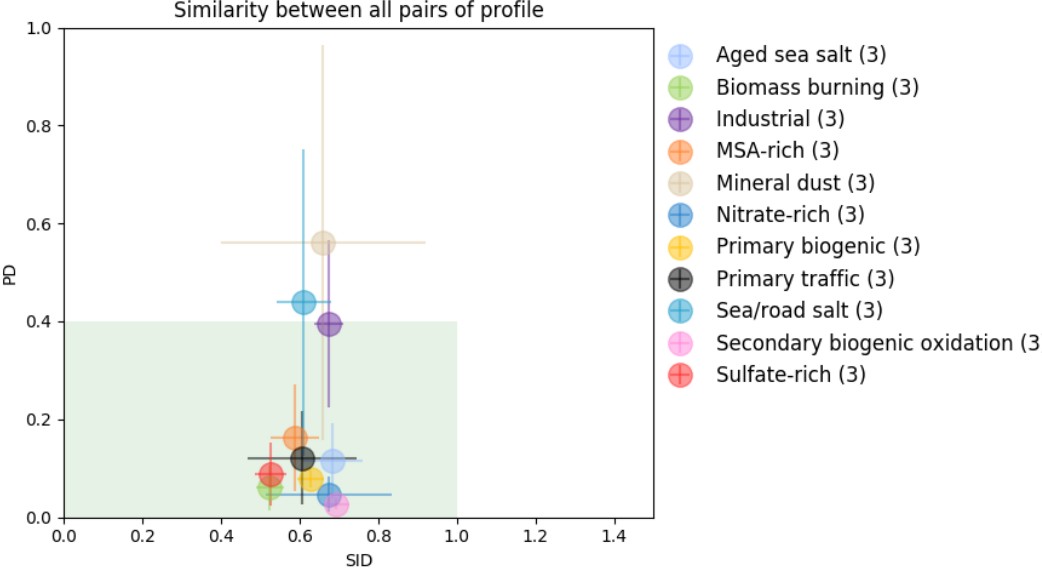

**Figure 11: Similarity plot of all chemical profiles in each site. The shaded area (in green) shows the acceptable range of the PD-SID**
**metric. For each point, the error bars represent the standard deviation of the 3 pairs of comparisons.**

Surprisingly, the sulfate-rich factor chemistry is one of the most stable profile, although its temporal contributions exhibits
high spatial variation, notably at CB compared to LF and Vif sites.
Finally, although the primary traffic factor showed a stable profile based on the similarity plot (PD-SID metric), it has to be
noted that, in the reference run (i.e., constrained), the specie concentrations are within the BS uncertainties for all species at
LF and CB sites, but outside the BS range for the Vif site (see Figure 12). Notably, the BS predicted higher contribution from
Cu, Fe, Sb and Sn, which are common tracers of tyre and brake wear, than the reference run. Additionally, the $Ca^{2+}$ is
overestimated in the reference run by a large amount, as well as the OC*, and, to a lesser extent, the reconstructed $PM_{10}$. Such
BS results indicate that, in Vif, the primary traffic factor is heavily influenced by this phenomenon on specific days, that has
led to an overestimation of the total $PM_{10}$ apportioned by this factor. Additionally, even at low concentrations, some terrestrial
elements (Al, As, Ti) and cellulose are present in the primary traffic factor. As a result, even if the primary traffic characteristic
of this factor is dominant, the influence of road dust re-suspension is not negligible for this factor in Vif.





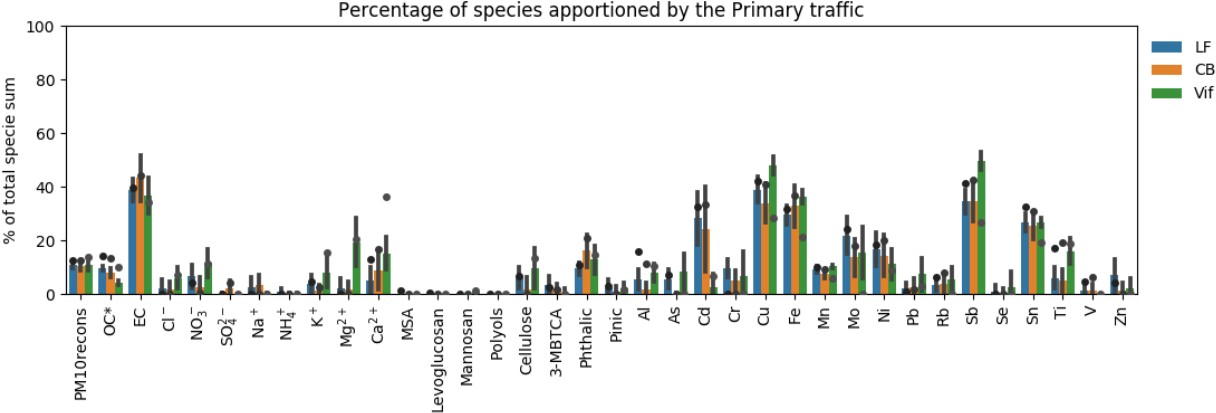


**Figure 12: Percentage (%) of each specie apportioned by the primary traffic factor (dots refer to the constrained run, bar plots refer to the mean and error bars refer to the standard deviation of the 100 BS).**


### 3.5.2. Comparison with other chemical profiles of PM$_{10}$ sources from a regional study

It is interesting to evaluate if a PMF study conducted at a city scale is leading to more similar source chemical profiles than a study using a database from a much larger area. Another question is whether PMF can produce more similar chemical source profiles with the help of additional organic tracers than a "classic" PMF run. Hence, the results obtained here are compared to those in the SOURCES program (Weber et al., 2019) for the 9 factors common in both studies (the secondary biogenic factor was not identified in the SOURCES program with the data sets not including its proper chemical tracers). This can be represented with the projection on a similarity plot of the distances between the factors for the 3 pairs of sites over the Grenoble basin, both for the "classic" and "orga" PMF. This is compared to the results from all possible pairs of sites within the 15 sites of the SOURCES study (distributed over France), mapped with a probability density function of similarities. Figure 13 presents these plots for the factors "primary traffic" and "sulfate-rich", the other factors being presented in the SI (S6).

It shows that in most instances, the PMF results obtained for the Grenoble basin deliver slightly closer chemical profiles, both for the PD distance (sensitive to major components) and the SID distance (sensitive to the global profile), than the studies across more distant sites. This is particularly the case when comparing the two urban sites (LF and CB) (cf., Figure 11). Some values out of the acceptable range remain for some factors (mineral dust, industrial), involving the differences between the urban and suburban sites, but are still fitting with the pattern obtained for the overall French sites. The addition of the organic tracers did not alter the source profiles of the commonly resolved PMF factors and, in fact, even enhanced it by further refining the other identified sources. This is predominantly seen in the MSA-rich factor, where some of the "classic" PMF results fell outside the acceptable range while the "orga" PMF are all in the acceptable range of the similarity plot. The "orga" PMF run for some factors such as primary biogenic, dust, and industrial factor also mostly yielded better PD and SID metrics (closer to the acceptable range) than the "classic" PMF run.

597





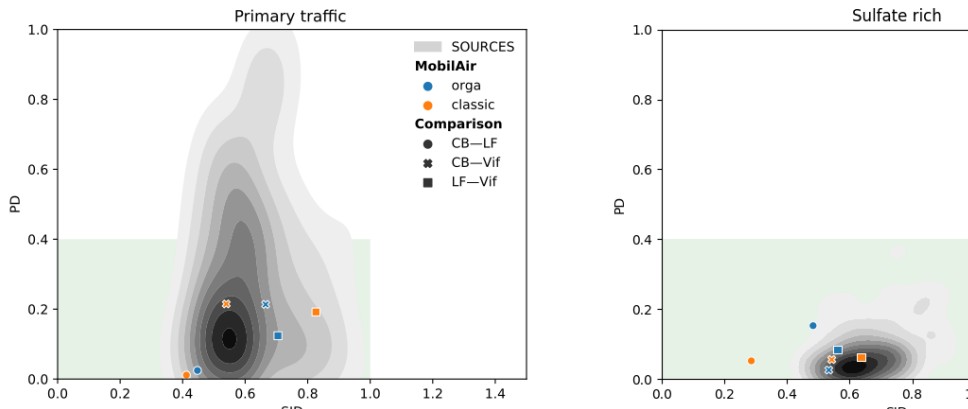

**Figure 13: Similarity plots for the factor "Primary traffic" and "Sulfate-rich" for the pairs of sites formed in this study (Mobil'Air), compared to the probability density function of similarities obtained for the 15 French sites of the SOURCES program.**

### 3.6 Improvement of the identified sources with the new organic tracers

In order to comprehensively apportion PM$_{10}$ sources, the very large unknown portion of OM, especially in the secondary fraction, needs to be properly identified. Most source apportionment studies only use standard input variables including OC, EC, ions, and metals. However, these species alone are insufficient to describe the complexity of the organic matter, making it a challenge to apportion sources from the OM fraction and their formation processes (i.e. primary or secondary origin) (Srivastava et al., 2018a). Only a small number of studies have used organic tracers to apportion SOA in PM using PMF, and even these studies usually have limited number of samples, number of tracers, and/or identified sources (Feng et al., 2013; Shrivastava et al., 2007; Srivastava et al., 2018b). A few of these studies have proposed to estimate SOC contributions from the sum of OC loadings in the secondary inorganic (nitrate- and sulfate-rich) factor (Hu et al., 2010; Ke et al., 2008; Lee et al., 2008; Pachon et al., 2010; Yuan et al., 2006) or from water-soluble OC and humic-like substances (Qiao et al., 2018). Some have estimated the contributions of biogenic SOA from the oxidation products of isoprene, alpha-pinene, and beta-caryophyllene (Heo et al., 2013; Kleindienst et al., 2007; Miyazaki et al., 2012; Shrivastava et al., 2007; Wang et al., 2012). In fact, the high contributions of biogenic SOA during warmer periods, that could range from 20-60% (Heo et al., 2013; Miyazaki et al., 2012; Wang et al., 2012; Zhang et al., 2010), found by other PMF studies is also consistent with our findings. Wang et al. (2017b) also highlighted the importance of biogenic SOA tracers as it significantly impacts the source apportionment results, particularly in areas with strong SOA contributions. Although applied on a small sample size, an interesting technique assimilating polar SOA tracers and primary organic aerosol (POA) tracers was performed by Hu et al. (2010) resulting to two identified SOA factors that are mixed with: 1) secondary inorganics and biomass burning, 2) green waste and biomass burning. Hence, with the appropriate uncertainties, the SOA tracers can be a practical way, possibly even necessary, to estimate SOA contributions (Feng et al., 2013) especially in urban areas (Wang et al., 2012).





Our study demonstrated that the use of organic tracers aided an effective source-specific approach that clearly identified major sources of SOA in PM$_{10}$ such as MSA-rich and secondary biogenic oxidation sources. The potential influence of anthropogenic emissions on some sources was also observed through the contribution of one of the organic tracers used, phthalic acid. The sufficient number of samples (n>125 for each site) in our study have also maintained the statistical robustness of the solutions obtained from filter-based measurements. The stability of the organic tracers also resulted to homogeneous chemical profiles which allowed seamless identification of uncommonly resolved sources such as primary biogenic, secondary biogenic oxidation, and MSA-rich. Although the addition of cellulose did not emerge as a separate biogenic factor, it provided an option to scrutinize the difference in terms origin of the primary biogenic sources across sites. Overall, the organic tracers have further refined the contribution of other identified sources by taking in consideration the SOA portion of PM$_{10}$ that would have otherwise mixed with other sources and have facilitated an innovative approach to improve the apportionment of PM$_{10}$ sources.

## 4 Conclusions

A fine-scale source apportionment of PM$_{10}$ in different urban typologies (background, pedestrianized hyper-center, and peri-urban) in a small scale area (<15 km) was performed using PMF 5.0. Additional organic tracers (MSA, cellulose, 3-MBTCA, pinic acid, and phthalic acid) were used to supplement the standard input variables. An 11-factor optimal solution was found for each of the three urban sites, including primary traffic, nitrate-rich, sulfate-rich, industrial, biomass burning, aged sea salt, sea/road salt, mineral dust, primary biogenic, secondary biogenic oxidation, and MSA-rich sources. The results from previously reported PMF studies in Grenoble (Srivastava et al., 2018b; Weber et al., 2019) were confirmed by the findings in this study particularly the long-term stability of regional source emissions 5 years apart. The PMF solution obtained with the additional organic tracers resulted to:

1. the improvement of PM$_{10}$ mass closure and the exploration of appropriate input variable uncertainties;
2. the re-assignment of the bulk sulfate-rich factor contribution to more descriptive secondary aerosol sources in the atmosphere;
3. the clear identification of commonly unresolved sources in the SOA fraction (e.g., primary biogenic, traced by the polyols and cellulose; secondary biogenic oxidation, traced by 3-MBTCA and pinic acid; and MSA-rich, traced by the MSA) in different urban typologies;
4. the decreased uncertainties, for both BS and DISP error estimates, that further strengthened confidence in the PMF solution;
5. the increased knowledge of the stability of the chemical profiles of the factors, that could be a key when using them in further large-scale analysis or modeling;

The 3 sites comparison at a local scale:



6. highlights very similar profiles and temporal evolution in the factor contributions at a conurbation scale (such as the Grenoble basin);

7. allows the determination of local heterogeneities in a small scale area;

8. pointed out some difficulties to disentangle the secondary inorganics sources ($NO_3^-$ and $SO_4^{2-}$) and some mixing between both species may occur.

Overall, an enhanced and fine-scale source profile of $PM_{10}$ was obtained in the Grenoble basin. The trend observed in the MSA-rich, secondary biogenic oxidation, and primary biogenic factors showed the extent of this phenomenon suggesting importance of the contribution of biogenic sources, both primary and secondary. The significant percentage attributed to SOA sources revealed the strong necessity of organic molecular tracers in fully discriminating the origins of $PM_{10}$ sources.

## Acknowledgements

This work is supported by the French National Research Agency in the framework of the "Investissements d'avenir" program (ANR-15-IDEX-02), for the Mobil'Air program. It also received support from the program QAMECS funded by ADEME (convention 1662C0029), and from LCSQA and French Ministry of Environment for part of the analyses for the Les Frenes site within the CARA program. Chemical analysis on the Air-O-Sol facility at IGE was made possible with the funding of some of the equipment by the Labex OSUG@2020 (ANR10 LABX56). The PhD of SW is funded by ENS Paris. The internship of T Cañete is taking place within the Erasmus exchange program. Finally, the authors would like to kindly thank the dedicated efforts of many people from Atmo-AuRA at the sampling sites, and in the lab at IGE (A Vella, C Vérin, C Voiron) for collecting and analysing the samples, respectively.

## Authors contributions

GU, and JLJ designed the atmospheric chemistry part of the Mobil'Air program. SM and CT supervised the sampling at the 3 sites for Atmo AuRA. OF is the head of the CARA program that allows the collection of samples from Les Frênes site. VJ set up the analytical techniques for polyols, sugars, and cellulose. TC performed the cellulose analyses. LJSB and SW processed the data. SW developed some of the tools and ideas for in-depth PMF analysis. LJSB, SW, wrote the paper. JLJ and GU revised the original draft. All authors reviewed and edited the manuscript.

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
