# Peer review of "Disparities in particulate matter $(PM_{10})$ origins and oxidative"

_Atmospheric Chemistry and Physics, 2020_

## Referee Comment (RC1) · Anonymous Referee #1 · 21 Jan 2021

The manuscript presents one year datasets of PM10 at three city sites in France. This is the first part of their study: compositions and source apportionment. The authors analyzed four kinds of pollutants: carbonaceous aerosol, inorganic ions, trace metals and organic biomarkers to discuss their possible source categories and spatial-temporal variations. The authors used PMF to quantitatively apportion 11 distinct sources that were deeply described in the manuscript. The apportioned sources are mostly based on previous study results and therefore convincing. However, I don't think this is a high quality manuscript since there were too many long sentences and not easily to follow.

I suggest the authors add discussions on the seasonal variation of sources categories and make a careful revision of the sentence structure.

Specific comments: abstract: the authors should point out the contribution ratio of 11 factors to the total sources. (?%)

introduction: compared with the organic markers the authors analyzed, persistent organic pollutants, such as PAHs, n-alkanes have been widely used to trace specific sources in cities as they were potential mutagens.

2.1 PM10 sample collection: the reasons that choose these three sites are not substantial.

2.2 the QA/QC requirements for detecting carbonaceous pollutants and inorganic ions are not mentioned.

3 results and discussion: there could be pollution days during one year sampling. How about the variation of the source categories between normal and polluted days?

---

## Referee Comment (RC2) · Zongbo Shi (Referee) · 31 Jan 2021

This is a solid paper showing source apportionment results at three different sites in Grenoble based on inorganic and organic tracers. It is well presented. Uncertainties and limitations of PMF analyses are comprehensively investigated, which is excellent. Interpretation is usually well justified.

I recommend publication after minor revision.

No major concern is identified. Detailed comments below:

[Figure]

Understood that why you did not present the composition data in detail. But it would still be useful to present the mass closure (e.g., as pie chart). I would prefer to see it in the main paper.

Abstract: Homogenous and heterogenous sources are not widely used terms; how these are defined?

Line 123: Please justify with more evidence why 1.8 OM/OC ratio is chosen, particularly for the particular location.

Section 2.2 / 2.3: Please describe the blank correction procedures briefly, and the levels in field blanks in SI?

Line 137-141: what is the recovery of trace elements in a reference material? Al concentration really low - is the recovery low?

Line 184: define OC* in the main text

Line 213: I do not think you mean "source profile". It should be factor profile. Revise across the manuscript accordingly. Source profile refers to measured profile from source aerosols.

2.4.3: why started from 7 factors, e.g., not 5? Based on experience?

Line 243: rewrite – two "contribution" in the same sentence; high contribution of inorganic ions does not necessarily suggest long-range transport? Strong inversion, indicated later, could mean that the air is stagnant for some time in the region, making chemical conversion possible.

What is the altitude of the three sites? Similar ? This could have implications if the mixing layer height is low.

Figure 2: explain why sometimes there are large spikes (differences) for some of the observed species.

Line 276: explain what caused the episode of phthalic acid

Figure 3: specify a correlation between what and what

Table 2: Aged sea salt should have sulfate or nitrate; it appears that there is an artificial split in the profile? Looking at the factor profile (line 149 in SI), you do not have contribution from nitrate. If the figure is presented as concentration in the left Y axis and contribution in the right Y axis, then it might be easier to see. OC is very high but it does appear in any of the factors as a tracer, which is odd. Look at SI (line 94), you do have contribution from OC

Table S2: Al is extremely low – Are you confident of this data? Could it be due to the very low extraction efficiency? Or may be there is no / little mineral dust ?

SI – Line 168: More in the winter suggesting road salt. But can you explain the spikes – sometimes they appear at all sites but sometimes not.

SI Figure S3.6 – Why the time series for CB is not consistent with those of the other two sites? It appears that the trends are opposite. Again, I would really like to see the concentration (in addition to the contribution) in the left figures. This factor seems to have a bit of everything with a high contribution from Se – does this suggest a strong coal combustion signature?

Figure S3.8 – Do the time series follow MSA concentrations? Can you explain the spikes that appear in some but not other sites? The large contribution to PM10 mass is puzzling. Perhaps you could go back to your original data to check for potential mistakes? And definitely should check for the mass closure.

Fig. S3.10- Factor profile are remarkably similar, which is good. Why Vif shows a rather different time series?

Figure S5.2 – If you remove the few outliers then the correlation might be very different. So perhaps you should check your data quality for those datapoints or find a potential reason why these are outliers? For example, meteorological factors?

3.2.1: It would be useful to provide some more discussions on the origins of some of the factors such as sulfate rich factor, as you did for the other factors. Perhaps not for all but at least for some of the more tricky ones (e.g., sulfate rich)

Line 459-467: the source of Ca2+; it is often used as a tracer for construction dust but there may not be a lot of construction activities in the city. Are local soils rich in carbonate? And why the loading of Ca2+ in the primary traffic is high? Is it from the resuspended dust or is it from the primary engine emissions?

Line 499-301: some explanations are given here but this could be enhanced. Could meteorology play a role (if one is at a high altitude so mixing layer height plays a role)? Could there be local sources that are present at one site but not the others?

Line 505 – yes, this is why I suggested above to identify the reasons behind the outliers, and show the correlation with and without such outliers.

Line 507 – Ok, this is sensible but please can you analyse the meteorological data to support this hypothesis?

Line 509 – Can you combine the sulfate and nitrate factors and correlate them? This might help with the argument of the possible "inability" for PMF to separate?

Fig. 9 is interesting and there got to be some reasons behind this (see above suggestions)

Line 520 – 529 : the analysis here is very interesting; why dust at Vif appears to be from Spain but not the others is interesting. Can you explain this by the topography, including the altitude of the sampling sites?

---

## Author Comment (AC1) · 3 Mar 2021

**Disparities in particulate matter (PM$_{10}$) origins and oxidative potential at a city-scale (Grenoble, France) - Part I: Source apportionment at three neighbouring sites**

Authors' response

We would like to thank the referees for their time to evaluate our manuscript and for their positive and constructive feedbacks, which helped improving the quality of the paper. Our point-by-point response to the comments are presented below with the referee comments in black, our answers in red, and changes in the revised version of the manuscript are printed in blue.

Anonymous referee #1: Received and published 21 January 2021

The manuscript presents one year datasets of PM10 at three city sites in France. This is the first part of their study: compositions and source apportionment. The authors analysed four kinds of pollutants: carbonaceous aerosol, inorganic ions, trace metals and organic biomarkers to discuss their possible source categories and spatial-temporal variations. The authors used PMF to quantitatively apportion 11 distinct sources that were deeply described in the manuscript. The apportioned sources are mostly based on previous study results and therefore convincing. However, I don't think this is a high quality manuscript since there were too many long sentences and not easily to follow. I suggest the authors add discussions on the seasonal variation of sources categories and make a careful revision of the sentence structure.

Reply: We appreciate the comments. As suggested by the referee, the text was revised in order to shorten the longest sentences and make it clearer and simpler.
Regarding the seasonality of PM source's relative contributions, this is now provided in the supplementary information (S3) along with the apportionment of OC per season.

**Figure S**2.1 **Seasonal contributions of the PMF-resolved sources to PM$_{10}$**

[Figure]

Specific comments: abstract: the authors should point out the contribution ratio of 11 factors to the total sources. (?%)

The relative contribution of each source over the 3 sites was added in the abstract as follows: An 11-factor solution was obtained in all sites including commonly identified sources from primary traffic (13%), nitrate-rich (17%), sulfate-rich (17%), industrial (1%), biomass burning (22%), aged sea salt (4%), sea/road salt (3%), and mineral dust (7%), and the newly found
sources from primary biogenic (4%), secondary biogenic oxidation (10%), and MSA-rich (3%).
introduction: compared with the organic markers the authors analyzed, persistent organic
pollutants, such as PAHs, n-alkanes have been widely used to trace specific sources in cities as
they were potential mutagens.
Reply: To address the referee's concern, we have added some references that used organic
markers (specifically persistent organic markers) and PMF for source apportionment (Bullock
et al., 2008; Wang et al., 2017a; Yan et al., 2017; Zhang et al., 2010). However, the tracers used
in our study for the PMF purpose should not be confused with important species associated
with health impacts, like POPs and PAHs. PAHs, although widely used, are emitted by all types
of combustion sources making it less straightforward.
We would also like to point out that we used additional fit-for-purpose tracers that can be
obtained using simpler and more targeted techniques to allow easier apportionment of PM
sources (like 3-MBTCA and cellulose). However, a previous study at one of the same sites
previously made use of some of PAHs and PAH derivatives. The results of the two studies are
compared in the supplementary information.
2.1 PM10 sample collection: the reasons that choose these three sites are not substantial.
Reply: One of the main goals of this study is to investigate the variation of source identification
and contribution within a small geographic scale and according to various site typology. The
intricate topography and seasonality of particulate air pollution in the city of Grenoble (France)
makes it an ideal location to explore these variabilities. To address the referee's comment, we
added this information and the altitude of each sampling site as follows: The topography within
the Grenoble basin and seasonality of particulate air pollution in the city makes it an ideal
location to explore both the small- and large-scale variabilities of PM sources. Within this study,
a PM10 sampling campaign was conducted in the Grenoble area at three sites selected to
represent various urban typologies, including: Les Frênes (LF, urban background site, 214
masl), Caserne de Bonne (CB, urban hyper-center, 212 masl), and Vif (peri-urban area, 310
masl).
2.2 the QA/QC requirements for detecting carbonaceous pollutants and inorganic ions are not
mentioned.
Reply: Thank you for this comment. Filter sampling as well as chemical analyses have been
performed following the recommendations of related EN standards (i.e., EN 12341, EN 14902,
EN 16909, EN 16913). Moreover, quality control of the chemical speciation analyses includes
chemical mass closure exercises.
It should also be noticed that our group successfully participates in regular inter-laboratory
comparison exercise for organic and elemental carbon (OC and EC) within ACTRIS and in
EMEP for ions analysis.
All of this was added in the revised manuscript as follows: The procedures for filter sampling
and chemical analyses have been performed following the recommendations of related EN
standards (i.e., EN 12341, EN 14902, EN 16909, EN 16913) (Favez et al., 2021). Moreover,
quality control of the chemical speciation analyses includes chemical mass closure as presented in the supplementary information (S2).  It should also be noted that our group successfully
participates in regular inter-laboratory comparison exercises for OC and EC within ACTRIS
and in EMEP (European Monitoring and Evaluation Programme) for ions analysis.

Limit of quantifications, determined using field blank measurements, are also now given in the
SI:

Action:

Table S3. The average of the field blanks of the campaign used to set the quantification limit (QL) of the species

| Specie | OC | EC | MSA | $Cl^-$ | $NO_3^-$ | $SO_4^{2-}$ | $Na^+$ | $NH_4^+$ | $K^+$ | $Mg^{2+}$ | $Ca^{2+}$ |
|---|---|---|---|---|---|---|---|---|---|---|---|
| Unit | $\mu g/m^3$ | $\mu g/m^3$ | $ng/m^3$ | $ng/m^3$ | $ng/m^3$ | $ng/m^3$ | $ng/m^3$ | $ng/m^3$ | $ng/m^3$ | $ng/m^3$ | $ng/m^3$ |
| QL | 0.06 | 0.01 | 0.06 | 9.29 | 17.16 | 11.00 | 16.54 | 23.34 | 3.10 | 1.04 | 5.23 |

| Specie | Arabitol | Mannitol | Levoglucosan | Mannosan | Cellulose | 3-MBTCA | Phthalic acid | Pinic acid |
|---|---|---|---|---|---|---|---|---|
| Unit | $ng/m^3$ | $ng/m^3$ | $ng/m^3$ | $ng/m^3$ | $ng/m^3$ | $ng/m^3$ | $ng/m^3$ | $ng/m^3$ |
| QL | 0.74 | 0.74 | 0.59 | 0.74 | 10.00 | 0.20 | 0.03 | 0.08 |

3 results and discussion: there could be pollution days during one-year sampling. How about
the variation of the source categories between normal and polluted days?

Reply: As stated in the introduction, the main point of this paper was to investigate the potential
of the PMF approach to deconvolute sources at small spatial scale, and to evaluate the benefit
to use innovative tracers. However, the reviewer is right when pointing out that the relative
source contribution highly differs for "normal" and "polluted" days. In Grenoble, polluted days
are often observed during wintertime due to severe thermal inversion or during spring due to
ammonium nitrate events.

To address the referee's concern, we have compared the relative source contribution between
low-pollution days (daily $PM_{10}$ concentration $\leq$ 30 µg m$^{-3}$) and more polluted days (daily $PM_{10}$
concentration > 30 µg m$^{-3}$) in the supplementary information (Figure S2.3). The 30 µg/m$^{-3}$
threshold was set arbitrarily, roughly corresponding to the 10%-highest daily concentrations
within the dataset available for the study. The absolute contributions of the sources to the total
$PM_{10}$ for each of these days are also given in the table below. We clearly see the impacts of
both the biomass burning and the nitrate-rich sources, but also the effect of thermal inversion
keeping the emissions from road traffic near the surface.

**Figure S2.3 Seasonal contributions of the PMF-resolved sources to PM$_{10}$ during normal days ($\leq$30 µg m$^{-3}$) and polluted**
**days (>30 µg m$^{-3}$)**

[Figure]

| Grenoble Les Frenes | | | | | | |
|---|---|---|---|---|---|---|
| | 2017-11-22 | 2017-11-24 | 2017-12-04 | 2017-12-07 | 2017-12-22 | 2017-12-25 |
| Aged sea salt | 0,1 | 0,2 | 0,5 | 0,1 | 0,7 | -0,1 |
| Biomass burning | 12,1 | 9,3 | 9,2 | 12,5 | 15,2 | 13,7 |
| Industrial | 0,2 | 0,4 | 0,1 | 0,7 | 0,5 | 0 |
| Mineral dust | 3,5 | 4,7 | 0,6 | 0,1 | -0,2 | -0,3 |
| MSA-rich | 0,2 | 0,3 | 0,4 | 0,2 | 0,3 | 0,2 |
| Nitrate-rich | 2 | 2,8 | 18,6 | 12,2 | 13,7 | 10,8 |
| Primary biogenic | 0,3 | 0,3 | 0 | 0,1 | 0,1 | 0,1 |
| Primary traffic | 13,3 | 13 | 3 | 4,2 | 7,9 | 0,2 |

This table provides the contribution (in µg m$^{-3}$) of the sources at each of the sites during some of the high level PM$_{10}$ episodes.

Referee #2 (Zongbo Shi, z.shi@bham.ac.uk ): Received and published 31 January 2021

This is a solid paper showing source apportionment results at three different sites in Grenoble based on inorganic and organic tracers. It is well presented. Uncertainties and limitations of

PMF analyses are comprehensively investigated, which is excellent. Interpretation is usually well justified. I recommend publication after minor revision. No major concern is identified.

Detailed comments below:

Reply: We appreciate the positive feedback.

Understood that why you did not present the composition data in detail. But it would still be useful to present the mass closure (e.g., as pie chart). I would prefer to see it in the main paper.

Reply: We appreciate this comment. The discussion on the PMF mass closure for PM$_{10}$ are in section 3.2.1. The authors deem that further discussion of the PM composition would not add any further new information and would be redundant with the other discussions in the paper.

However, we are now providing a pie chart of the mean annual compositions at each site and a scatterplot comparison of the PMF-reconstructed PM$_{10}$ and observed PM$_{10}$ in the supplementary information (S2).

**Figure S1.1 Percentage composition of PM$_{10}$**

[Figure]

Mass closure of main PM components

**Figure S1.2: A scatterplot comparison of the PMF-reconstructed PM$_{10}$ and observed PM$_{10}$**

[Figure]

Abstract: Homogenous and heterogenous sources are not widely used terms; how these are defined?

We also made the meaning clearer about "homogeneous" and "heterogeneous" sources: The PD-SID metric was used to determine whether the profiles attributed to a given source can be considered as homogeneous (i.e. with similar chemical profiles over the 3 sites) - or heterogeneous - at the city scale.

Line 123: Please justify with more evidence why 1.8 OM/OC ratio is chosen, particularly for the particular location.

Reply: Such a ratio commonly used in the literature for urban sites in Europe. Also, it was mentioned that this choice has been notably based on two references (Favez et al., 2010; Putaud et al., 2010). In fact, the study by Favez et al. (2010) was conducted at the same site LF ("Les Frenes"), where a mean OC-to-OM conversion factor of 1.78±0.17 is obtained from the comparison between aerosol mass spectrometer (AMS) and low pressure cascade impactor (LPI) measurements.

Section 2.2 / 2.3: Please describe the blank correction procedures briefly, and the levels in field blanks in SI?

Reply: Field blanks were collected throughout the sampling period to blank-correct all samples accordingly. The blanks were obtained with filters following all stages of sample preparation
and analysis. Blanks are analysed for all of the species measured. The average of the field blanks
was removed from the values obtained for the real samples and defined as the Quantification
Level (QL) (see answer to referee #1 for the values). This is now mentioned in the
supplementary information (Table S3).

Line 137-141: what is the recovery of trace elements in a reference material? Al concentration
really low - is the recovery low?

Reply: The recovery varies in the range 80-110 % according to the element. The data are not
corrected for specific recovery. The numbers for Al were in µg m$^{-3}$, we have corrected and
updated Table S2. We acknowledge that the concentrations are rather low, but they are in the
range of previous measurements in the area, and in the low range observed in Alastuey et al.
(2016) for European cities.

Alastuey, A., Querol, X., Aas, W., Lucarelli, F., Pérez, N., Moreno, T., Cavalli, F., Areskoug,
H., Balan, V., Catrambone, M., Ceburnis, D., Cerro, J. C., Conil, S., Gevorgyan, L., Hueglin,
C., Imre, K., Jaffrezo, J.-L., Leeson, S. R., Mihalopoulos, N., Mitosinkova, M., O'Dowd, C. D.,
Pey, J., Putaud, J.-P., Riffault, V., Ripoll, A., Sciare, J., Sellegri, K., Spindler, G., and Yttri, K.
E.: Geochemistry of PM10 over Europe during the EMEP intensive measurement periods in
summer 2012 and winter 2013, Atmos. Chem. Phys., 16, 6107–6129,
https://doi.org/10.5194/acp-16-6107-2016, 2016.

Table S2. Annual average of PM$_{10}$ mass concentrations and chemical compositions (in µg m$^{-3}$
or ng m$^{-3}$) at all sites, and individual urban sites in the Grenoble basin.

| Species | Unit | Mean [Q1, Q3] | | | |
|---|---|---|---|---|---|
| | | All sites | CB (urban hyper-center) | LF (urban background) | Vif (peri-urban) |
| PM10recons | µg/m³ | 14.4 [8.0, 17.8] | 16.0 [8.8, 20.3] | 14.2 [8.1, 17.2] | 13.1 [7.3, 16.5] |
| OC* | | 3.95 [2.28, 5.0] | 4.14 [2.43, 5.28] | 3.95 [2.28, 4.73] | 3.75 [2.12, 4.49] |
| EC | | 1.01 [0.46, 1.32] | 1.18 [0.57, 1.5] | 1.12 [0.53, 1.35] | 0.73 [0.34, 0.85] |
| Cl- | | 0.12 [0.01, 0.1] | 0.16 [0.02, 0.15] | 0.08 [0.01, 0.08] | 0.1 [0.0, 0.08] |
| NO3- | | 2.02 [0.48, 2.11] | 2.55 [0.67, 3.16] | 1.78 [0.51, 1.7] | 1.72 [0.36, 1.7] |
| SO42- | | 1.48 [0.81, 1.89] | 1.58 [0.89, 2.0] | 1.53 [0.87, 1.97] | 1.33 [0.69, 1.74] |
| Na+ | | 0.17 [0.07, 0.2] | 0.2 [0.08, 0.24] | 0.15 [0.06, 0.19] | 0.15 [0.06, 0.18] |
| NH4+ | | 0.85 [0.3, 0.89] | 0.99 [0.31, 1.11] | 0.81 [0.32, 0.81] | 0.75 [0.27, 0.79] |
| K+ | | 0.15 [0.07, 0.18] | 0.16 [0.08, 0.19] | 0.15 [0.07, 0.17] | 0.13 [0.06, 0.17] |
| Mg2+ | | 0.02 [0.01, 0.02] | 0.02 [0.01, 0.03] | 0.02 [0.01, 0.02] | 0.02 [0.01, 0.02] |
| Ca2+ | | 0.32 [0.13, 0.44] | 0.36 [0.13, 0.52] | 0.31 [0.12, 0.38] | 0.3 [0.13, 0.42] |
| MSA | | 0.02 [0.01, 0.03] | 0.03 [0.01, 0.03] | 0.02 [0.01, 0.03] | 0.02 [0.01, 0.03] |
| Levoglucosan | | 0.3 [0.02, 0.42] | 0.25 [0.02, 0.35] | 0.28 [0.02, 0.42] | 0.36 [0.02, 0.47] |
| Mannosan | | 0.03 [0.0, 0.04] | 0.03 [0.0, 0.04] | 0.03 [0.0, 0.05] | 0.04 [0.0, 0.05] |
| Polyols | | 0.04 [0.01, 0.06] | 0.04 [0.01, 0.06] | 0.04 [0.01, 0.06] | 0.05 [0.01, 0.07] |
| Cellulose | | 0.08 [0.02, 0.12] | 0.13 [0.07, 0.17] | 0.05 [0.02, 0.08] | 0.06 [0.01, 0.09] |
| 3-MBTCA | ng/m³ | 9.13 [1.75, 12.92] | 9.8 [1.83, 13.18] | 8.5 [1.72, 11.89] | 9.09 [1.69, 13.18] |
| Phthalic_acid | | 3.54 [1.8, 4.02] | 3.5 [1.82, 4.13] | 3.88 [1.88, 4.68] | 3.24 [1.78, 3.82] |

| | | | | |
|---|---|---|---|---|
| Pinic_acid | 6.61 [2.3, 7.83] | 5.36 [1.65, 7.21] | 5.25 [2.48, 6.66] | 9.22 [2.94, 11.28] |
| Al | 62.67 [19.6, 68.7] | 62.26 [22.41, 73.59] | 65.58 [21.95, 68.43] | 60.19 [16.82, 63.54] |
| As | 0.33 [0.14, 0.39] | 0.41 [0.16, 0.47] | 0.37 [0.17, 0.48] | 0.23 [0.11, 0.27] |
| Cd | 0.07 [0.02, 0.09] | 0.08 [0.02, 0.1] | 0.07 [0.02, 0.09] | 0.05 [0.01, 0.06] |
| Cr | 1.65 [0.61, 1.73] | 2.27 [0.79, 2.23] | 1.61 [0.7, 1.79] | 1.05 [0.61, 1.01] |
| Cu | 8.5 [3.82, 9.8] | 11.59 [5.17, 13.27] | 8.79 [4.08, 10.24] | 5.09 [2.72, 6.18] |
| Fe | 215.26 [91.41, 270.23] | 241.66 [104.95, 290.45] | 248.53 [112.83, 299.27] | 155.64 [68.3, 184.7] |
| Mn | 9.0 [2.73, 9.36] | 11.73 [3.38, 11.77] | 7.19 [2.63, 8.31] | 8.03 [2.21, 7.09] |
| Mo | 0.59 [0.19, 0.65] | 0.8 [0.25, 0.92] | 0.63 [0.21, 0.67] | 0.35 [0.13, 0.41] |
| Ni | 0.91 [0.37, 1.07] | 1.18 [0.5, 1.4] | 0.92 [0.39, 1.12] | 0.63 [0.3, 0.75] |
| Pb | 4.42 [1.52, 5.01] | 5.73 [2.0, 7.23] | 4.84 [1.72, 5.75] | 2.69 [1.15, 3.06] |
| Rb | 0.45 [0.21, 0.58] | 0.48 [0.25, 0.6] | 0.44 [0.21, 0.57] | 0.41 [0.18, 0.58] |
| Sb | 1.31 [0.33, 0.93] | 1.71 [0.46, 1.33] | 1.53 [0.4, 1.26] | 0.69 [0.22, 0.51] |
| Se | 0.39 [0.23, 0.5] | 0.43 [0.27, 0.54] | 0.41 [0.26, 0.53] | 0.32 [0.18, 0.43] |
| Sn | 2.26 [1.41, 2.63] | 2.6 [1.55, 3.13] | 2.45 [1.49, 2.96] | 1.73 [1.28, 2.03] |
| Ti | 3.81 [1.6, 4.95] | 4.11 [1.8, 5.57] | 3.83 [1.68, 5.08] | 3.49 [1.38, 4.32] |
| V | 0.48 [0.16, 0.62] | 0.51 [0.19, 0.62] | 0.52 [0.16, 0.65] | 0.42 [0.13, 0.52] |
| Zn | 20.27 [6.09, 21.82] | 26.11 [8.18, 28.63] | 23.58 [8.69, 24.41] | 11.11 [3.64, 12.07] |

Line 184: define OC* in the main text

Reply: The meaning of OC* has been explained in detail in the supplementary information (Eq.
S2) together with other specifics in the PMF methodology. This is now also stated in the revised
manuscript: In order to avoid double counting of carbon mass, OC* was calculated as the
difference between total OC and the quantity of C atoms contained in the different organic
markers include in the PMF input data matrix (as detailed in Eq. S2).

Line 213: I do not think you mean "source profile". It should be factor profile. Revise across
the manuscript accordingly. Source profile refers to measured profile from source aerosols.

Reply: Yes, we agree that the referee has a point and we appreciate this comment. We
acknowledge the fact that we are not measuring PM chemistry directly at their source, hence
the more accurate term would be "factors". However, the term "sources" is easier to understand
than "factors" and it is widely used like that in all the literature about PMF studies. In our
studies, we are however often keeping the term "source" when the chemical profile is clearly
associated with a type of source (i.e. "biomass burning"), and the term "factor" when it is not
("nitrate-rich").

2.4.3: why started from 7 factors, e.g., not 5? Based on experience?

Reply: Indeed, based from experience, with the given set of species in the Grenoble basin, or
more broadly in France (cf references below), we are not expecting less than 6 or 7 factors (biomass burning, mineral dust, secondary inorganics (one or two factors), primary biogenic,
road traffic, salt (aged/fresh)) (Bonvalot et al., 2016, 2019; Favez et al., 2021; Salameh et al.,
2018; Srivastava et al., 2018; Waked et al., 2014; Weber et al., 2019).
Similarly, inter-comparison of receptor models often reports more than 7 factors when using
the traditional/basic set of species (without organic acid or cellulose) (Belis et al., 2020). The
addition of new proxies, with very different contributions and temporal distribution, should lead
to the identification of even more factors.
References:

Belis, C. A., Pernigotti, D., Pirovano, G., Favez, O., Jaffrezo, J.-L., Kuenen, J., Denier van Der Gon, H., Reizer, M., Riffault, V., Alleman, L. Y., Almeida, M., Amato, F., Angyal, A., Argyropoulos, G., Bande, S., Beslic, I., Besombes, J.-L., Bove, M. C., Brotto, P., Calori, G., Cesari, D., Colombi, C., Contini, D., De Gennaro, G., Di Gilio, A., Diapouli, E., El Haddad, I., Elbern, H., Eleftheriadis, K., Ferreira, J., Vivanco, M. G., Gilardoni, S., Golly, B., Hellebust, S., Hopke, P. K., Izadmanesh, Y., Jorquera, H., Krajsek, K., Kranenburg, R., Lazzeri, P., Lenartz, F., Lucarelli, F., Maciejewska, K., Manders, A., Manousakas, M., Masiol, M., Mircea, M., Mooibroek, D., Nava, S., Oliveira, D., Paglione, M., Pandolfi, M., Perrone, M., Petralia, E., Pietrodangelo, A., Pillon, S., Pokorna, P., Prati, P., Salameh, D., Samara, C., Samek, L., Saraga, D., Sauvage, S., Schaap, M., Scotto, F., Sega, K., Siour, G., Tauler, R., Valli, G., Vecchi, R., Venturini, E., Vestenius, M., Waked, A. and Yubero, E.: Evaluation of receptor and chemical transport models for PM10 source apportionment, Atmospheric Environ. X, 5, 100053, https://doi.org/10.1016/j.aeaoa.2019.100053, 2020.

Bonvalot, L., Tuna, T., Fagault, Y., Jaffrezo, J.-L., Jacob, V., Chevrier, F. and Bard, E.: Estimating contributions from biomass burning, fossil fuel combustion, and biogenic carbon to carbonaceous aerosols in the Valley of Chamonix: a dual approach based on radiocarbon and levoglucosan, Atmospheric Chem. Phys., 16(21), 13753–13772, https://doi.org/10.5194/acp-16-13753-2016, 2016.

Bonvalot, L., Tuna, T., Fagault, Y., Sylvestre, A., Mesbah, B., Wortham, H., Jaffrezo, J.-L., Marchand, N. and Bard, E.: Source apportionment of carbonaceous aerosols in the vicinity of a Mediterranean industrial harbor: A coupled approach based on radiocarbon and molecular tracers, Atmos. Environ., 212, 250–261, https://doi.org/10.1016/j.atmosenv.2019.04.008, 2019.
Favez, O., Weber, S., Petit, J.-E., Alleman, L. Y., Albinet, A., Riffault, V., Chazeau, B., Amodeo, T., Salameh, D., Zhang, Y., Srivastava, D., Samaké, A., Aujay-Plouzeau, R., Papin, A., Bonnaire, N., Boullanger, C., Chatain, M., Chevrier, F., Detournay, A., Dominik-Sègue, M., Falhun, R., Garbin, C., Ghersi, V., Grignion, G., Levigoureux, G., Pontet, S., Rangognio, J., Zhang, S., Besombes, J.-L., Conil, S., Uzu, G., Savarino, J., Marchand, N., Gros, V., Marchand, C., Jaffrezo, J.-L. and Leoz-Garziandia, E.: Overview of the French Operational Network for In Situ Observation of PM Chemical Composition and Sources in Urban Environments (CARA Program), Atmosphere, 12(2), 207, https://doi.org/10.3390/atmos12020207, 2021.

Salameh, D., Pey, J., Bozzetti, C., El Haddad, I., Detournay, A., Sylvestre, A., Canonaco, F., Armengaud, A., Piga, D., Robin, D., Prevot, A. S. H., Jaffrezo, J.-L., Wortham, H. and Marchand, N.: Sources of PM2.5 at an urban-industrial Mediterranean city, Marseille (France): Application of the ME-2 solver to inorganic and organic markers, Atmospheric Res., 214, 263–274, https://doi.org/10.1016/j.atmosres.2018.08.005, 2018.

Srivastava, D., Tomaz, S., Favez, O., Lanzafame, G. M., Golly, B., Besombes, J.-L., Alleman, L. Y., Jaffrezo, J.-L., Jacob, V., Perraudin, E., Villenave, E. and Albinet, A.: Speciation of organic fraction does matter for source apportionment. Part 1: A one-year campaign in Grenoble (France), Sci. Total Environ., 624, 1598–1611, https://doi.org/10.1016/j.scitotenv.2017.12.135, 2018.

Waked, A., Favez, O., Alleman, L. Y., Piot, C., Petit, J.-E., Delaunay, T., Verlinden, E., Golly, B., Besombes, J.-L., Jaffrezo, J.-L. and Leoz-Garziandia, E.: Source apportionment of PM$_{10}$ in a north-western Europe regional urban background site (Lens, France) using positive matrix factorization and including primary biogenic emissions, Atmospheric Chem. Phys., 14(7), 3325–3346, https://doi.org/10.5194/acp-14-3325-2014, 2014.

Weber, S., Salameh, D., Albinet, A., Alleman, L. Y., Waked, A., Besombes, J.-L., Jacob, V., Guillaud, G., Mesbah, B., Rocq, B., Hulin, A., Dominik-Sègue, M., Chrétien, E., Jaffrezo, J.-L. and Favez, O.: Comparison of PM$_{10}$ Sources Profiles at 15 French Sites Using a Harmonized Constrained Positive Matrix Factorization Approach, Atmosphere, 10(6), 310, https://doi.org/10.3390/atmos10060310, 2019.

Line 243: rewrite – two "contribution" in the same sentence; high contribution of inorganic ions does not necessarily suggest long-range transport? Strong inversion, indicated later, could mean that the air is stagnant for some time in the region, making chemical conversion possible. What is the altitude of the three sites? Similar? This could have implications if the mixing layer height is low.

Reply: Thank you very much for this comment. As per suggestion of the referee, the first "contribution" word was removed. The altitudes of each site was also added in Line 102. We agree that the accumulation due to inversion may also be a phenomenon leading to the increase of secondary inorganic aerosols (SIA), as exemplified by one winter episode in these series. We added this hypothesis in the sentence. However, in most instances, synchronous episodes of SIA (notably ammonium nitrate) can be detected over very large fraction of the French territory.

This was followed by contributions from the major inorganic species (NH$_4^+$, NO$_3^-$, and SO$_4^{2-}$), suggesting strong influence from secondary inorganic aerosol (SIA) that are generally associated with long-range transport of pollutants, or, in some instances with the occurrence of a local thermal inversion within the Grenoble basin.

Frênes (LF, urban background site, 214 masl), Caserne de Bonne (CB, urban hyper-center, 212 masl), and Vif (peri-urban area, 310 masl).

Figure 2: explain why sometimes there are large spikes (differences) for some of the observed species.

Reply: Indeed, this is one of the main points of the paper—to determine which sources have a similar influence over the 3 sites and which sources do not (see section 3.4 and 3.5). Most of them are discussed in the article (organic acid notably) and here in the Authors' response, but some others remain under more thorough investigation.

Line 276: explain what caused the episode of phthalic acid

Reply: We appreciate this comment, that pushed us into more investigations. On the days of
this short-term episode, there were also spikes in sulfate- and nitrate-rich sources. Our
hypothesis was that of an occurrence of a persistent fog episode over the Grenoble area where
mixed phase reaction in the droplets could happen. This is supported by observation from a
webcam from a high location with a view over the Grenoble basin. We clearly see this strong
fog staying for 4 days (from Feb 22 to 26). See
https://www.skaping.com/grenoble/bastille?archives=MTUxOTM4NjA2MA-YQ for Feb.
$22^{th}$, 2018. We added some discussion in the revised manuscript as follows: An extensive
discussion on the formation processes of anthropogenic SOA in high concentration events was
already provided in Srivastava et al. (2018b). However, this new observation brings in the
hypothesis that these processes may take place specifically due to heterogeneous chemistry
when associated with fog episodes.

Figure 3: specify a correlation between what and what

Reply: Thank you for this comment. This figure shows the correlation between sites in terms
of their PM composition. For clarity, we updated the legend that now reads as follows:

Figure 3: Concentration time series Pearson correlation coefficient of $PM_{10}$ and its chemical
composition between LF and CB (LF-CB), LF and Vif (LF-Vif), and CB and Vif (CB-Vif).

Table 2: Aged sea salt should have sulfate or nitrate; it appears that there is an artificial split in
the profile? Looking at the factor profile (line 149 in SI), you do not have contribution from
nitrate. If the figure is presented as concentration in the left Y axis and contribution in the right
Y axis, then it might be easier to see. OC is very high but it does appear in any of the factors as
a tracer, which is odd. Look at SI (line 94), you do have contribution from OC

Reply:
We fully agree with the reviewer that the aged sea-salt should have at least some nitrate or
sulfate in it. The figure requested by the reviewer (absolute concentration per microgram of
PM) is given below for the urban background (LF) site. We clearly see the impact of nitrate
and, to a lesser extent, sulfate in this factor.
We agree that this representation per microgram is important and useful, and we closely
monitored these during the source identification process. However, we decided not to show
these figures for all factors, in order to stay as concise as possible, thinking that the chemical
profile and temporal evolutions with error estimates (provided in S3) were deemed sufficient
for the objectives of this study.
Regarding OC: OC* is not defined as a specific tracer of any source, since it can come from a
very large array of the sources determined in this study; as such, it is not listed in Table 2.
However, OC is known to be a component of many sources and for example, a biomass burning
factor without OC would be highly suspicious. Similarly, nitrate or sulfate are not specific
tracers of aged salt, hence these are not listed in Table 2. The contributions of the sources to

OC is now provided in Figure S.2.2.

[Figure]

Relative mass concentration for the Aged sea salt factor at Grenoble Les Frenes (UB) with the boxplots representing the bootstrap uncertainties.

Table S2: Al is extremely low – Are you confident of this data? Could it be due to the very low
extraction efficiency? Or maybe there is no / little mineral dust?
Reply: Please see answer above.
SI – Line 168: More in the winter suggesting road salt. But can you explain the spikes –
sometimes they appear at all sites but sometimes not.
Reply: Indeed, this factor is identified as "sea and road salt". Most of the spikes during winter
appear simultaneously, due to resuspension of road salt. The road salting being discontinuous
in time explains the spikes observed for this factor. The biggest differences in the spikes are
between CB and LF when compared to Vif. It is fully in agreement with the proximity of CB
and LF, being similarly influenced and the distance of Vif (more than 15 km away).
SI Figure S3.6 – Why the time series for CB is not consistent with those of the other two sites?
It appears that the trends are opposite. Again, I would really like to see the concentration (in
addition to the contribution) in the left figures. This factor seems to have a bit of everything
with a high contribution from Se – does this suggest a strong coal combustion signature?
Reply: This is a very interesting comment, since the sulfate rich factor (like the nitrate-rich one)
is generally a poorly defined factor, and it should be one goal of the PMF research area to better
characterize its associated sources and/or processes. We believe that it is one strong
achievement of this study to show that by adding 3-MBTCA and pinic acids as tracers of
biogenic SOA, we can split the sulfate-rich factor (and this one only) getting a "new" biogenic-
SOA source. In much the same way, our work using isotopic measurements of N was able to
provide some added value at the delineation of the nitrate-rich factor (Favez et al., 2021).
It should be noted that there is virtually no coal combustion sources in France. Atmospheric
selenium can indeed come from a variety of anthropogenic emissions (see the discussions on
the sources in the references quoted below) and is consistently found in the sulfate-rich factors
in France (Weber et al. 2019). Further, despite potential biogenic sources of DMS as indicated
by a systematic MSA-rich factor in our PMF, most of the sulfate should be from anthropogenic origins if one considers classical MSA-to-sulfate ratios from the DMS oxidation products.
Coming to the specific results of this study, we agree that the temporal trend of the sulfate-rich
factor in CB is not similar with the one in LF and Vif (Figure S3.6 in the SI). In the two latter
sites, the nitrate-rich factor apportioned a small amount of sulfate (around ~20% of the total
sulfate), but almost no sulfate is present in the nitrate-rich factor of CB. Conversely, more
nitrate is apportioned in the sulfate-rich factor of CB compared to LF and Vif (see figure below).

[Figure]

This resulted to a lower contribution of the nitrate-rich source during the spike of spring 2018
for CB compared to LF and Vif (see Figure S3.6 in the SI). An important fraction of this spike
is then explained by the sulfate-rich factor, instead of the nitrate-rich. In the end, this is a case
of potential mixing between these two factors in both LF and Vif.
A further indication of the potential mixing between the 2 factors, that would be different
according to the sites is given by the following figure, suggested by the reviewer, showing the
correlations between sites of the sums of the major ions (sulfate, nitrate, ammonium) and of the
PM included in each of these 2 factors. The correlations for these 4 plots are really high,
indicating that the sum of the 2 factors are very similar between all sites for these 4
concentrations, but become variable if single factors are considered. It is out of reach of this
work to determine if this is a limitation of the PMF approach, or if there are some processes
leading to real differences. However, Figure 9 of the main paper is showing that this difference
between sites is really robust for the sulfate factor (with CB being apart from the 2 other sites),
with the estimation of uncertainties given by bootstrapping 500 runs. We have added a figure
in the supplementary information (Figure S5.3) and a discussion in the revised manuscript that
reads: A further indication of a potential mixing between the sulfate- and nitrate-rich factors is
presented in Figure S5.3. In this figure, the total mass concentration of PM and major ions
($SO_4^{2-}$, $NO_3^-$, and $NH_4^+$) were compared between sites when the sulfate- and nitrate-rich factors
were combined. Strong correlations between sites were found indicating similarity of such
concentrations in secondary sources.  It is out of scope of this work to determine if this is a

 limitation of the PMF approach, or if there are some processes leading to real differences.

[Figure]

[Figure]

*Relative mass concentration for the Sulfate rich factor at Grenoble Caserne de bonne (CB) with the boxplots representing the bootstrap uncertainties.*

[Figure]

**Figure S5.3. Scatterplot comparison of total mass concentration of PM and major ions (SO$_4^{2-}$, NO$_3^-$, and NH$_4^+$) between**
**sites when the sulfate- and nitrate-rich factors were combined**

Arimoto, R., Duce, R.A., Savoie, D.L. *et al.* Trace elements in aerosol particles from Bermuda
and Barbados: Concentrations, sources and relationships to aerosol sulfate. *J Atmos*
*Chem* **14,** 439–457 (1992). https://doi.org/10.1007/BF00115250

Santiago, A., Longo, A., Ingall, E. *et al.*. Characterization of Selenium in Ambient Aerosols
and Primary Emission Sources. *Environmental Science & Technology* 48 (16), 8988-8994
(2014 ) DOI: 10.1021/es500379y

Figure S3.8 – Do the time series follow MSA concentrations? Can you explain the spikes that
appear in some but not other sites? The large contribution to PM10 mass is puzzling. Perhaps
you could go back to your original data to check for potential mistakes? And definitely should
check for the mass closure.

Reply: This factor is definitely driven by the MSA concentrations, MSA being the primary
tracer for it. For now, we cannot explain the origin of the few spikes that happen in the series.
The MSA concentrations for these sites are well within range of previous measurements in
continental areas (Golly et al., 2019; Crippa et al., 2013; Weber et al., 2019). We also provided
a comparison between the PMF-reconstructed PM$_{10}$ and the observed PM$_{10}$ from TEOM in the supplementary information (Figure S2.7):

**Figure S2.7 Comparison between PMF-reconstructed PM$_{10}$ and observed PM$_{10}$ from TEOM in µg m$^{-3}$**

[Figure]

Crippa, M., El Haddad, I., Slowik, J.G., DeCarlo, P.F., Mohr, C., Heringa, M.F., Chirico, R.,
Marchand, N., Sciare, J., Baltensperger, U., Prévôt, A.S.H. Identification of marine and
continental aerosol sources in Paris using high resolution aerosol mass spectrometry. J.
Geophys. Res. Atmospheres 118, 1950–1963. https://doi.org/10.1002/jgrd.50151, 2013.

Golly, B., Waked, A., Weber, S., Samake, A., Jacob, V., Conil, S., Rangognio, J., Chrétien, E.,
Vagnot, M.-P., Robic, P.-Y., Besombes, J.-L. and Jaffrezo, J.-L.: Organic markers and OC
source apportionment for seasonal variations of PM2.5 at 5 rural sites in France, Atmospheric
Environment, 198, 142–157, https://doi.org/10.1016/j.atmosenv.2018.10.027, 2019.

Fig. S3.10- Factor profile are remarkably similar, which is good. Why Vif shows a rather
different time series?

Reply: It is important to note that even if there is a large body of species commonly found in
this factor, this factor presents important uncertainties with regards to the total PM mass
apportioned. Due to these uncertainties, the industrial source appears, in fact, heterogeneous
(mostly in the PD metric, due the amount of OC and EC) as discussed in section 3.5.1. In fact,
the temporal contributions of this source do not present strong correlation between sites (see
Figure 8 in the main text). We also present here the example of the industrial factor of CB in
µg/µg of PM:

[Figure]

*Figure 1: Industrial factor of CB in µg/µg of PM.*

We clearly see here the important uncertainties and even some "impossible" values where the
mass of OC* or Fe are greater than the total mass of PM$_{10}$ in some outputs of the bootstrap runs,
due to extremely low or even null amount of PM that is sometimes determined by the PMF in
this factor. Then, if we can say with strong confidence that an "industrial" factor was found,
strong uncertainties are still attached to this result and its concentration series should be
interpreted with caution.

Figure S5.2 – If you remove the few outliers then the correlation might be very different. So
perhaps you should check your data quality for those datapoints or find a potential reason why
these are outliers? For example, meteorological factors?

Reply: The illustration of this phenomenon is the whole point of this figure. These few points
are days having different conditions, hence the discussion in Line 505 and the back
trajectory/PSCF analysis on the two different possible sources of mineral dust (very local, hence
influencing only CB and LF, or long-range transport, influencing the three sites
simultaneously).

3.2.1: It would be useful to provide some more discussions on the origins of some of the factors
such as sulfate rich factor, as you did for the other factors. Perhaps not for all but at least for
some of the more tricky ones (e.g., sulfate rich)

Reply: We agree that the sulfate-rich factor presents an interesting case, showing interesting
points in this study, and it was discussed in Line 510.

Line 459-467: the source of Ca2+; it is often used as a tracer for construction dust but there
may not be a lot of construction activities in the city. Are local soils rich in carbonate? And why
the loading of Ca2+ in the primary traffic is high? Is it from the resuspended dust or is it from
the primary engine emissions?

Reply: Ca$^{2+}$ in an urban environment is not a really good tracer, since it can come from several
sources: it can be from construction activities (as there are always some sort of construction
going on, and notably the construction of a highway around Grenoble), but also from global
resuspended dust from many activities, from biomass burning, and also from resuspended dust
with traffic (hence the presence in the traffic factor). It should be noted that previous studies comparing measurements at LF and near a close highway 2 km apart showed an increment of 34% near the highway, giving an idea of the resuspended dust with traffic (Charron et al.,2019). The grounds in the city are mainly paved. Bare soils are rare. It may be less the case in Vif with more natural soils. The Grenoble valley is a glacial valley with soils made of a variety of origin but with a large share of carbonate (limestone and marl).

We added some discussion in the revised manuscript as follows: It is important to note that $Ca^{2+}$ in an urban environment can come from several sources such as construction activities and global resuspended dust from various activities (from biomass burning and traffic). Previous studies comparing measurements at LF and a site close to a highway (2 km apart) showed a 34% increment of this factor near the highway, supporting the influence of resuspended dust with traffic (Charron et al., 2019).

Line 499-301: some explanations are given here but this could be enhanced. Could meteorology play a role (if one is at a high altitude so mixing layer height plays a role)? Could there be local sources that are present at one site but not the others?

Reply: Indeed, all sites are in close proximity in terms of altitude (Les Frênes (LF, urban background site, 214 masl), Caserne de Bonne (CB, urban hyper-center, 212 masl), and Vif (peri-urban area, 310 masl)). This paper did not focus on the role of meteorological factors, instead this paper discusses the diversity of the sources, both in terms of fine scale variability in temporal distribution and chemical profiles, whilst also taking into account the typology.

Line 505 – yes, this is why I suggested above to identify the reasons behind the outliers, and show the correlation with and without such outliers.

Reply: Please see above discussion about the potential mixing of sulfate-rich and nitrate-rich.

Line 507 – Ok, this is sensible but please can you analyse the meteorological data to support this hypothesis?

Reply: We do not have meteorological data, however we can provide a visualization of the visibility status over the Grenoble basin, please see https://bastille-grenoble.fr/webcam/?lang=en.  We are also in the process of investigating specific fog episodes with series of measurements over the Grenoble basin.

Line 509 – Can you combine the sulfate and nitrate factors and correlate them? This might help with the argument of the possible "inability" for PMF to separate?

Reply: We thank the referee for this idea. Please see discussion about these 2 factors above.

Fig. 9 is interesting and there got to be some reasons behind this (see above suggestions)

Reply: Thank you for this comment. Please see above discussion on this topic.

Line 520 – 529 : the analysis here is very interesting; why dust at Vif appears to be from Spain
but not the others is interesting. Can you explain this by the topography, including the altitude
of the sampling sites?
Reply: Thank you very much for this comment. Local sources near LF and CB, such as
construction works, strongly affect these two sites, but not Vif. Only large scale phenomena
such as Saharan dust episodes affect the three site simultaneously (hence the Figure S.5.2, with
some days with very high similarities for mineral dust when comparing CB and Vif).
Indeed, there may be some specificity in the atmospheric dynamics in the valley near the
surface. Vif is in the south valley with air canalized by valley and katabic flows in a south to
north direction. The air flow in other sites are more perturbed by the flow coming from the
eastern valley. However, synoptic circulation affects the three sites simultaneously. This
phenomenon is also reported in the reference below. This point is addressed in the revised
manuscript as follows: This is indicative of two regimes for mineral dust, with differences due
to some specificity in the atmospheric dynamics in the valley near the surface. To investigate it
further, a potential source contribution function (PSCF) analysis of the mineral dust factor for
the Vif and CB sites was performed in order to assess the origin of air masses of this factor
Figure 10). For the Vif site, the main origin is Spain with well-defined air flow canalized by the
valley and katabic flows in a south to north direction (a phenomenon also reported in Largeron
and Staquet (2016)), whereas the origin for CB is not as well-defined. These PSCF pattern tends
to indicate that the sources of the mineral dust factor present a strong local component for the
urban sites (CB and LF being very similar), while the origin of the mineral dust factor in Vif
appears to be mainly affected by long-range transport of dust only.
Source: Largeron, Y. and Staquet, C. (2016) The Atmospheric Boundary Layer during Winter
time Persistent Inversions in the Grenoble Valleys. Front. EarthSci.4:70. doi:
10.3389/feart.2016.00070